



# A daily and 500m coupled evapotranspiration and gross primary production product across China during 2000-2020

Shaoyang He[1,2], Yongqiang Zhang[1*], Ning Ma[1], Jing Tian[1], Dongdong Kong[3], Changming Liu[1]

[1]Key Laboratory of Water Cycle and Related Land Surface Processes, Institute of Geographic Sciences and Natural Resources Research, CAS, Beijing 100101, China
[2]University of Chinese Academy of Sciences, Beijing 100040, China
[3]Department of Atmospheric Science, School of Environmental Studies, China University of Geosciences, Wuhan 430074, China

*Correspondence to*: Yongqiang Zhang (zhangyq@igsnrr.ac.cn)

**Abstract.** Accurate high-resolution actual evapotranspiration (ET) and gross primary production (GPP) information is essential for understanding the large-scale water and carbon dynamics. However, substantial uncertainties exist in the current ET and GPP datasets in China because of insufficient local ground measurements used for model constrain. This study utilizes a water-carbon coupled model, Penman-Monteith-Leuning Version 2 (PML-V2), to estimate 500m ET and GPP at a daily scale. The parameters of PML-V2(China) were well-calibrated against observations of 26 eddy covariance flux towers across nine plant functional types in China, indicated by Nash-Sutcliffe Efficiency (*NSE*) of 0.75 and Root Mean Square Error (*RMSE*) of 0.69 mm $d^{-1}$ for daily ET respectively, and *NSE* of 0.82 and *RMSE* of 1.71 g C $m^{-2}$ $d^{-1}$ for daily GPP. The model estimates get a small *bias* of 6.28% and a high *NSE* of 0.82 against water-balance annual ET estimates across 10 major river basins in China. Further evaluations suggest that the newly developed product outperforms its global version and other typical products (MOD16A2, SEBAL, GLEAM, MOD17A2H, VPM, and EC-LUE) in estimating both ET and GPP. Moreover, PML-V2(China) accurately monitors the intra-annual variations in ET and GPP in the croplands with a dual-cropping system. Using the new data showed that, over the last 20 years, the annual GPP and water use efficiency experienced a significant (*p* < 0.001) increase (8.51 g C $m^{-2}$ $yr^{-1}$ and 0.02 g C $mm^{-1}$ $H_2O$ $yr^{-1}$, respectively), but annual ET showed a non-significant (*p* > 0.05) increase (0.65 mm $yr^{-1}$). This indicates that vegetation in China exhibits a huge potential for carbon sequestration with little cost in water resources. The PML-V2(China) product provides a great opportunity for academic communities and various agencies for scientific studies and applications, freely available at http://dx.doi.org/10.11888/Terre.tpdc.272389 (Zhang and He, 2022).

## 1 Introduction

Terrestrial evapotranspiration (ET) and photosynthesis (or gross primary productivity, GPP) are indispensable processes in hydrological and carbon cycles at global and regional scales. Forming the second largest water flux after precipitation in the terrestrial hydrological cycle, ET is the sum of plant transpiration ($E_c$), evaporation from the soil ($E_s$), and canopy



evaporation from precipitation interception ($E_i$). Photosynthesis rate, tightly coupled to $E_c$ by leaf stomata, is a key indicator of plant growth and provides food and fibre for human society.

In recent decades, numerous studies have been carried out to map ET at regional, continental, and global scales. Particular credit goes to remote sensing (RS)-based models that provide diagnostic ET estimates with a relatively high spatiotemporal

continuity and reasonable biophysical significance. For example, Miralles et al. (2011b) and Martens et al. (2017) developed a global and daily ET product employing the Global Land Evaporation Amsterdam Model 3.0a (hereafter GLEAM) based on the Priestley-Taylor (P-T) equation. Although GLEAM does a good job in temporal resolution, it has a coarse spatial resolution of 0.25°. Another widely used semi-empirical formula is Penman-Monteith (PM) equation. Mu et al. (2011) generated the MOD16A2 ET with 500m and 8-day resolutions based on PM equation as one global product of Moderate

Resolution Imaging Spectroradiometer (MODIS). Leuning et al. (2008) developed the Penman-Monteith-Leuning (PML) model that described the physical characteristics of canopy-soil water loss by improving the surface conductance ($G_s$) formulations and Zhang et al. (2010) estimated PML-based ET at 0.05° and 8-day resolution across the Australian continent. Cheng et al. (2021) produced a 1-km and daily ET dataset across China by the Surface Energy Balance Algorithm for Land (SEBAL), but only evaluated the SEBAL product at eight EC sites of three land cover types.

Apart from methods and products for the ET estimated above, numerous approaches have been used to estimate GPP, such as the enzyme kinetic (process-based) models (Houborg et al., 2013; Arain et al., 2006; Grant et al., 2005; Hanson et al., 2004; Medvigy et al., 2009), light use efficiency (LUE) principle (Liu et al., 2003; Turner et al., 2003; Yuan et al., 2007; Running et al., 2015; Zhang et al., 2017; Zheng et al., 2020), statistical methods (Potter et al., 1993; Hilker et al., 2008; Zhang et al., 2020b) and machine learning methods (Wolanin et al., 2019; Joiner et al., 2020; Huang et al., 2021). Among

them, the LUE principle is well known because of its simple structure, strong portability, and relatively high temporal cover of inputs. Running et al. (2015) recently updated the global GPP product of MODIS (hereafter MOD17A2H) at 500m and 8-day resolutions using the LUE principle. Zhang et al. (2017) and Zhang et al. (2021) also mapped global GPP for 2000 - 2019 dubbed Vegetation Photosynthesis Model GPP V20 (hereafter VPM) with the same spatiotemporal resolution as MOD17A2H using an improved LUE model adding the energy absorbed by chlorophyll. Zheng et al. (2020) generated a

global GPP dataset at 0.05° and 8-day intervals by a revised LUE model (hereafter EC-LUE) integrating the atmospheric $CO_2$ concentration.

Although significant efforts have been put into estimating ET and GPP, there are barely any coupled products available in China, which meet the requirement of high temporal ($\leq$ 1 day) and high spatial ($\leq$ 500m) resolutions simultaneously that is necessary to detect variations of the eco-hydrological cycle in diverse and large areas for a long term precisely (Table 1). For

instance, products with low temporal resolutions cannot detect subtle seasonal changes in areas seriously affected by human activities and in arid regions, such as irrigated farmland with a dry climate (Bodner et al., 2015). On the other hand, products with high temporal resolutions like GLEAM can monitor the diurnal variability of ET, but their low spatial resolutions limit the effectiveness in fine-scale environment applications (Gevaert and García-Haro, 2015).



Secondly, the phenomenon of ignoring the water-carbon coupling process frequently appearing in the existing products has brought systematic errors. The photosynthesis and transpiration are coupled by the plant stomatal control on both water and carbon exchange between the land ecosystem and the atmosphere (Xiao et al., 2013; Zhang et al., 2019). As indicated in Table 1, MODIS ET and MODIS GPP products are independent of each other, and cannot ensure similar biophysical characteristics of vegetation in the same place. Furthermore, using ET and GPP from different products can lead to large uncertainty in analysing the interaction between ET and GPP such as water use efficiency of ecosystem (WUE is the ratio of GPP to ET). In that case, it is necessary to build a coupled ET and GPP model considering the water-carbon coupling process. Zhang et al. (2019) developed the second generation of PML (i.e., PML-V2 model) that estimates $G_s$ using a water-carbon coupled model and mapped global ET and GPP at 500m and 8-day resolutions in 2002-2017.

More importantly, previous studies utilized sparse ground observations in China (as shown in Table 1) that covered few terrestrial ecosystems for models' calibration and validation, resulting in improper input parameters, and making it difficult to obtain more reliable estimates of ET and GPP in diverse landcover types (Heinsch et al., 2006; Sjöström et al., 2013). Although the eddy-covariance (EC) flux sites have provided consecutive measurements of water and carbon fluxes since the early 1990s (Xiao et al., 2013; Wofsy et al., 1993; Baldocchi et al., 2001), the EC observed data in China remain much sparser than those in North America and Europe, and most of them are not publicly available, impeding a national scale constraint of RS-based models for improving ET and GPP estimates (Villarreal and Vargas, 2021; Chu et al., 2017). For instance, GLEAM which only employed 8 EC sites over China overestimates ET at a large scale, especially for evergreen needleleaf forest, evergreen broadleaf forest and mixed forest (Li et al., 2018). Therefore, the uncertainty in estimating ET and GPP is large, and it requests sufficient EC flux sites to calibrate and validate ET and GPP models for better local and regional applications.

In addition, understanding the spatial and temporal pattern of ET and GPP is particularly important for China, the largest contributor to the absolute growth of greenhouse gases that directly induces global warming over the past decade (Minx et al., 2021). While China owns huge carbon sequestration potential of terrestrial ecosystems for slowing accumulation of atmospheric carbon dioxide and mitigating climate change. Additionally, given China's water shortage, it is crucial to clearly understand water budgets and transportation (Ma et al., 2020). Therefore, it becomes vital to estimate GPP and ET accurately across China under changes in climate and land cover types.

Facing the above challenges, this study utilizes a water-carbon coupled and remote sensing-based model, PML-V2 that is constrained against the most comprehensive observations in China (i.e. 26 EC observations across nine plant functional types (PFTs)), to generate daily and 500m ET and GPP data product from 2000 to 2020. We then test whether the newly developed product for China outperforms PML-V2(global) and other mainstream products (i.e., MOD16A2, SEBAL, GLEAM, MOD17A2H, VPM, and EC-LUE), and investigate spatial pattern and general trend over time in ET and its components ($E_c$, $E_i$, $E_s$), GPP and WUE in 2001-2020 across China.

The novelties of this study mainly include:

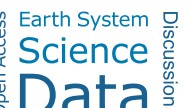

(i) Observation data from 26 EC flux stations in nine PFTs across China are employed for constraining the PML-V2 calibration for estimating ET and GPP;

(ii) The country-specific meteorological forcing, i.e., the CMFD, is used to drive the PML-V2 in China, which is more accurate than those forcings extracted from global forcing products;

(iii) The PML-V2(China) product is generated with a daily resolution, compared to the previous global product with a temporal resolution of 8-day; and

(v) Improving intra-annual ET and GPP dynamics for various ecosystems, particularly for the cropland ecosystem, which provides more accurate estimates and monitoring of agricultural water consumption, compared to other mainstream products.

## 2 Materials and methods

### 2.1 Description of the PML-V2 model

The PML-V2 model was employed in this study (Zhang et al., 2019). The detailed descriptions of PML-V2 are provided in Gan et al. (2018) and Zhang et al. (2019). Here we summarize the general modelling framework for readers to easily follow. PML-V2 derives ET by separately estimating its three components, including $E_c$, $E_s$ and $E_i$ (Eq.1). $E_s$ is calculated by a simple water availability model that requires only precipitation and equilibrium evapotranspiration (Zhang et al., 2010; Morillas et al., 2013), as shown in Eq.2. $E_i$ is estimated by a modified Gash model (Eq.3) that has been adopted widely as a rainfall interception function (Van Dijk and Bruijnzeel, 2001; Zhang et al., 2016). PML-V2 applies an improved Ball-Berry stomatal conductance model (Yu et al., 2004; Gan et al., 2018; Ball et al., 1987; Collatz et al., 1991), which couples $E_c$ calculated by PM equation (Eq.4) with GPP. Furthermore, GPP is constrained by vapor pressure deficit that shows the physiological significance of vegetation response, as shown in Eq.5 and Eq.6:

$$ET = E_c + E_s + E_i, \tag{1}$$

$$E_s = \frac{f \varepsilon A_s}{\varepsilon + 1}, \tag{2}$$

$$E_i = \begin{cases} f_v Prcp, Prcp < Prcp_{wet} \\ f_v Prcp_{wet} + f_{ER}(Prcp - Prcp_{wet}), Prcp \geq Prcp_{wet} \end{cases}, \tag{3}$$

$$E_c = \frac{\varepsilon A_c + (\rho c_p/\gamma) D_a G_a}{\varepsilon + 1 + G_a/G_c}, \tag{4}$$

$$GPP = A_g f(D_a), \tag{5}$$

$$f(D_a) = \begin{cases} 1, D_a \leq D_{min} \\ \frac{D_{max} - D_a}{D_{max} - D_{min}}, D_{min} < D_a < D_{max} \\ 1, D_a \geq D_{max} \end{cases}, \tag{6}$$

where $f$ is a unitless variable that is computed by a function of soil equilibrium evaporation and accumulated precipitation for each grid cell; $\varepsilon = s/\gamma$ (unitless), in which $s = de^*/dT$ is the curve slope relating saturation water vapor pressure to





temperature (kPa °C$^{-1}$) and $\gamma$ is psychrometric constant (kPa °C); the total available energy absorbed by surface is partitioned

by leaf area index (LAI, m$^2$ m$^{-2}$) into canopy absorption (A$_c$, MJ m$^{-2}$ d$^{-1}$) and soil absorption (A$_s$, MJ m$^{-2}$ d$^{-1}$); $\rho$ is the air

density (g m$^{-3}$); $c_p$ represents specific heat of air at constant pressure (J g$^{-1}$ °C$^{-1}$); $D_a$ is vapor pressure deficit (VPD) at the air

temperature; $G_a$ (m s$^{-1}$) is the aerodynamic conductance while $G_c$ (m s$^{-1}$) is the canopy conductance; $f_V$ is the area ratio

covered by intercepting leaves (unitless), $f_{ER}$ is the ratio of average evaporation rate over average precipitation (unitless) and

assumes that it does not vary between the storms; $Prcp$ is daily precipitation (mm d$^{-1}$); $Prcp_{wet}$ is rainfall rate of the

reference threshold if the vegetation canopy is wet (mm d$^{-1}$); $A_g$ (μmol m$^{-2}$ s$^{-1}$) is the gross assimilation rate and $f(D_a)$ is a

vapor pressure deficit constraint function where $D_{min}$ is the minimum threshold when there's no vapor pressure constraint

and $D_{max}$ the maximum threshold when closing plant stomata leads to non-assimilation.

## 2.2 Model input data

### 2.2.1 Remote sensing data

MODIS collections with 500m spatial resolutions from February 26, 2000 to December 31, 2020 (hereafter 2000-2020) are

used, which includes the LAI from MOD15A2H.006 (Myneni et al., 2015), the albedo from MCD43A3.006 product (Schaaf

and Wang, 2015), and the surface emissivity from MOD11A2.006 (Wan et al., 2015). The smoothed LAI inputs for PML-

V2(China) utilized the weighted Whittaker smoother with dynamic lambdas instead of a constant lambda to eliminate

missing or unreliable pixels due to noise contaminations by snow, shadow, cloud, etc, compared with the PML-V2(Global)

(Kong et al., 2019; Zhang et al., 2019). The improved LAI can better express peaks and seasonal changes. The albedo and

surface emissivity inputs were gap-filled by the linear interpolation of the nearest good quality points (Zhang et al., 2019). If

there were not enough good-quality points close to the point with missing value, it was filled by the historical averaged

values for the same grid. Besides, the PML-V2 model needs land cover types to accurately estimate ET and GPP in different

terrestrial ecosystems. Here we used the International Geosphere-Biosphere Program (IGBP) layer of MCD12Q1.006 land

cover product (Sulla-Menashe et al., 2019) during 2000-2020 since IGBP classification is annually continuous and has the

highest accuracy in China when compared with land cover products (Feng and Bai, 2019).

### 2.2.2 Meteorological data

The meteorological inputs of PML-V2 include specific humidity, air pressure, air temperature, wind speed, precipitation,

downward longwave radiation, downward shortwave radiation, and land surface temperature. In this study, the main

meteorological data came from the China Meteorological Forcing Dataset (CMFD) from 2000 to 2018 (He et al., 2020),

which has spatial and temporal resolutions of 0.1° and 3-hr, respectively. Generated through the fusion of five RS or

reanalysis datasets and 753 China Meteorological Administration stations, the CMFD dataset shows the best accuracy among

most available meteorological datasets, and is widely employed in hydrological and land surface modelling of China (Zhang

et al., 2019; Wang et al., 2020). The Global Land Data Assimilation System Version 2.1 (GLDAS-2.1) meteorological





forcing data with 0.25° and 3-hr resolutions (Beaudoing and Rodell, 2016) were used for 2019-2020, as the CMFD is not available during such a period. The land surface temperature during 2000-2020 was from the Land component of the fifth generation of European ReAnalysis (i.e., ERA5-Land; Muñoz-Sabater et al., 2021) with spatial and temporal resolutions of 0.1° and 1-hr, respectively. Note that the above meteorological data were first aggregated into the daily scale, followed by resampling into 500m by the bilinear interpolation method (Zhang et al., 2019). The atmospheric $CO_2$ concentration data

came from the National Oceanic and Atmospheric Administration. The pro-processing of remote sensing data and meteorological data for model inputs is summarised in Figure 1.

**2.3 ET and GPP from eddy-covariance observations**

We collated EC flux towers and automatic weather stations (AWSs) data from 26 sites across China (Fig. 2 and Table 2) for calibration and validation of PML-V2. These data came from the following sources: FLUXNET2015 (Pastorello et al., 2020),

the National Tibetan Plateau Data Center (Ma et al., 2020), the Heihe Integrated Observatory Network (Liu et al., 2011; Liu et al., 2018), and the Chinese Terrestrial Ecosystem Flux Research Network (ChinaFLUX) (Yu et al., 2006). These 26 sites encompass nine major PFTs in China, including two in evergreen needleleaf forests, one in evergreen broadleaf forests, one in mixed forests, one in open shrublands, one in savannas, eight in grasslands, three in wetlands, seven in croplands, and two in barren sparse vegetation. The observation variables, including air temperature, relative humidity, incoming shortwave

radiation, latent heat flux, sensible heat flux, and net ecosystem exchange, were collected from the interval of 0.5-hour or 1-hour to prepare for gap filling and flux partitioning (Reichstein et al., 2005; Wutzler et al., 2018). Considering certain gaps exist in the original half-hourly or hourly latent heat (LE) and net ecosystem exchange (NEE) fluxes data, we employed the marginal distribution sampling method (Reichstein et al., 2005) to fill these gaps using the station-observed air temperature, relative humidity, and solar radiation data. Subsequently, we partitioned NEE into gross carbon uptake (GPP) and respiration

of ecosystem according to the night-time-based method of Reichstein et al. (2005). Because any gap-filling of EC data may introduce extra uncertainties, we only used the days during which the percentage of the original observed and good-quality gap-filled data was no less than 60% in the present study. Note that the data of the sites from the ChinaFLUX (i.e., CN-CBF, CF-HBG_S01, CF-HBG_W01, CF-NMG, CF-QYF, and CF-YCA) and the FLUXNET2015 (i.e., CN-Cng, CN-Du2, and CN-HaM) have already been gap-filled by original data providers. Therefore, they were used directly in this study. Note that

while energy imbalance does exist in many EC sites, correcting such a problem may also introduce more uncertainties (Foken, 2008). Therefore, we used the observed LE directly in the present study.

**2.4 Basin-scale water balance-based evapotranspiration data**

The water balance method is generally regarded as a simple and accurate approach for calculating land evapotranspiration at the basin-scale (Liu et al., 2016). Here we used the water-balance-based evapotranspiration ($ET_{wb}$, mm) of 10 major basins

across China to evaluate PML-V2 ET estimates at a basin scale, that is,

$$ET_{wb} = Prcp - Q - TWSC, \qquad (7)$$





where $Prcp$, $Q$, and $TWSC$ (all with a unit of mm) are basin-wide precipitation, runoff, and change of terrestrial water storage at an annual scale, respectively. Among them, $Prcp$ and $Q$ are the annual values of ten major river basins in China, including the Hai, Huai, Liao, Northwest, Pearl, Songhua, Southeast, Southwest, Yangtze, and Yellow (Fig. 2), from the National Water Resources Bulletin, which is extensively used in water resources calculation (Miao et al., 2022) and assessment (Yang et al., 2004; Xie et al., 2018). $TWSC$ was quantified using three Gravity Recovery and Climate Experiment (GRACE) products (Landerer and Swenson, 2012; Landerer, F. 2021) including the NASA Jet Propulsion Laboratory, the GeoForschungsZentrum Potsdam and the Center for Space Research, which are available since April 2002 at a monthly scale. For reducing the uncertainties, this study used the mean values of these three products. We regarded the differences in the terrestrial water storage anomaly between two consecutive Decembers as the annual $TWSC$. Note that the $ET_{wb}$ was only calculated from 2003 to 2013 in the present study since the December values from GRACE was not available in 2014.

**2.5 Model calibration of PML-V2 model**

The Genetic algorithm (Holland, 1992; Konak et al., 2006) was used to calibrate the 11 parameters of the PML-V2 model for each PFT. Specifically, all EC-observed ET and GPP data within a PFT are used to minimize the following objective function ($F_{opt}$):

$$F_{opt} = 2 - NSE_{ET} - NSE_{GPP} = \frac{\sum_{i=1}^{N}(ET_{est}-ET_{obs})^2}{\sum_{i=1}^{N}(ET_{obs}-\overline{ET_{obs}})^2} + \frac{\sum_{i=1}^{N}(GPP_{est}-GPP_{obs})^2}{\sum_{i=1}^{N}(GPP_{obs}-\overline{GPP_{obs}})^2} \tag{8}$$

where $NSE_{ET}$ and $NSE_{GPP}$ are the Nash-Sutcliffe Efficiency of the daily ET and the daily GPP, respectively. The subscripts *est* and *obs* stand for the estimated and the observed, respectively. In this way, the calibrated parameter values for nine PFTs are illustrated in Table S1.

**2.6 Model validation and performance metrics**

The 'leave-one-out' cross-validation method was utilized to evaluate robustness of the PML-V2 model (Zhang et al., 2019). For each PFT, the data from one "ungauged" observation was excluded from the optimization while the data from all other observations at the same PFT were used for model calibration to obtain the simulated at the "ungauged" position. All nine PFTs were actualized in this way. Note that the PFT including EBF, MF, OSH, and SAV only has one ground site (Table 2). Therefore, it is appropriate to divide the data in each of the four sites into two sub-groups for cross-validation. The CF-CBF and the CF-HBG_S01 covering from 2003 to 2010, were divided into two sub-groups, each of which had 4 years: 2003-2006 and 2007-2010. While both the BNXJL and YJGRHG only covered one year and were divided into two sub-groups by a two-day time step, separately. After that, the daily estimates in the cross-validation mode were against the daily observation from the 26 stations to explore the model transferability from known observations to any location.

We assessed the performance of calibration and cross-validation of PML-V2 (and other seven mainstream ET and GPP products) against the observed sites or water-balance basins utilizing the following four metrics:





$$NSE_X = 1 - \frac{\sum_{i=1}^{N}(X_{est}-X_{obs})^2}{\sum_{i=1}^{N}(X_{obs}-\overline{X_{obs}})^2}, \tag{9}$$

$$R_X = \frac{\sum_{i=1}^{N}(X_{est}-\overline{X_{est}})(X_{obs}-\overline{X_{obs}})}{\sqrt{\sum_{i=1}^{N}(X_{est}-\overline{X_{est}})^2 \times \sum_{i=1}^{N}(X_{obs}-\overline{X_{obs}})^2}}, \tag{10}$$

$$RMSE_X = \sqrt{\frac{\sum_{i=1}^{N}(X_{est}-X_{obs})^2}{N}}, \tag{11}$$

$$Bias_X = \frac{\sum_{i=1}^{N}(X_{est}-X_{obs})}{N \times \overline{X_{obs}}}, \tag{12}$$

where $NSE$, $R$, $RMSE$, and $Bias$ are the Nash-Sutcliffe Efficiency, the correlation coefficient, the Root Mean Square Error, and the ratio of the difference between the estimated and the observed to the observed average. The subscript $X$ represents ET or GPP; the subscripts *est* and *obs* stand for the estimated and the observed, respectively.

**3 Results**

**3.1 Model calibration and model validation**

The simulated ET and GPP from the calibrated PML-V2(China) were first evaluated against EC measurements of 26 flux sites at a daily scale (Fig. 3). Overall, PML-V2(China) shows an excellent performance in estimating daily ET and daily GPP, as evidenced by the *NSE* (0.75 and 0.82, respectively), *R* (0.88 and 0.9, respectively), *RMSE* (0.69 mm d$^{-1}$ and 1.71 g C m$^{-2}$ d$^{-1}$, respectively), and *Bias* (-5.81% and -2.3%, respectively). For the mean values of each site, the simulated daily ET and daily GPP show higher *NSE* ($\geq 0.87$) and *R* ($\geq 0.93$) values (Fig. 3).

PML-V2(China) is only slightly degraded from calibration to cross-validation, indicated by slightly declined performance in ET and GPP (Fig. 3). For daily ET, the *NSE* and *R* values decrease by 0.06 and 0.04, respectively, from the calibration mode to the cross-validation mode. Correspondingly, the *RMSE* and *Bias* of ET in the cross-validation mode increase by 0.08 mm
d$^{-1}$ and 3.5%, respectively. For daily GPP, the *NSE* and *R* values in the cross-validation mode reduce by 0.11 and 0.06, respectively; the *RMSE* and *Bias* increase by 0.45 g C m$^{-2}$ d$^{-1}$ and 1.79%, respectively. A similarly slight degradation is applied to their site means. These results demonstrate that the PML-V2(China) is robust for estimating daily ET and daily GPP across large regions, and suitable for generating good quality daily ET and daily GPP data for China.

Figure 4 further summarises PML-V2(China) performance at 26 flux sites across nine PFTs. The estimates of ET and GPP
from the model calibration show high consistency with the EC-observed values in all terrestrial biomes. For daily ET (Fig. 4a), the *NSE* values vary from the range of 0.36 ~ 0.82, the *RMSE* 0.39 ~ 0.88 mm d$^{-1}$, and *Bias* -10.09% ~ -0.21%. For daily GPP (Fig. 4b), the ranges of statistical metrics become 0.41 ~ 0.91 for *NSE*, 0.3 ~ 3.19 g C m$^{-2}$ d$^{-1}$ for *RMSE*, and -10.52% ~ 3.26% for *Bias*. In terms of cross-validation, nine PFTs all showed slight declines in the statistical metrics when compared to those in the calibration mode. For daily ET, the declines in *NSE* values are less than 0.14 in most PFTs except BSV and ENF,
whose *NSE* decreased by 0.36 and 0.33, respectively. As expected, *RMSE* values all increased to some extent in all PFTs



(ranging from 0.002 to 0.305 mm d$^{-1}$) when compared with those in calibration mode. The *Bias* values in the cross-validation mode were almost identical to those in the calibration mode for most PFTs except WET and ENF of which the absolute value of *Bias* increased by 10.59% and 17.42%, respectively (Fig. 4a). Regarding daily GPP, the *NSE* values all degraded by less than 0.04 for most PFTs except BSV, GRA and WET where there exists 0.21 ~ 0.32 *NSE* degradation. In the meantime,

the declines in *R* values are all within 0.19. Regarding *RMSE*, the increases are particularly marginal for most PFTs except WET with an increase of 1.58 g C m$^{-2}$ d$^{-1}$. The *Bias* in the cross-validation mode is also similar to that in the calibration mode for most PFTs except for GRA and WET of which the absolute value *Bias* rose by 10.26% and 11.86%, respectively (Fig. 4b). The above PFT tests suggest that the present PML-V2(China), with parameter values being calibrated against 26 EC flux stations, does perform satisfactorily in estimating both ET and GPP across different PFTs in China.

To investigate the model performance at each EC site, this study also compares the variations in daily ET and GPP between PML-V2(China) in calibration mode and the EC observations (Fig. 5, Fig. 6, and Table S2). Overall, the estimates from PML-V2(China) show similar amplitude and phase to the EC observations, indicating that it performs well in capturing the seasonal phase of ET and GPP in most flux sites. From Figure 5, the PML-V2(China) ET reveals a single peak in each annual cycle for most flux sites, except those with double-cropping systems such as CF-YCA, DXZ, and GTZ cropland sites.

In particular, the ET values are consistent with observed data at the desert site (e.g., HZZHMZ and QZ-NAMORS) with *NSE* ranging from 0.41 to 0.48, indicating that model performs well in the sparse vegetation area. The measurements fluctuate higher than PML-V2(China) ET at peak areas for most sites. For the site mean ET, the difference values between PML-V2(China) and EC observations range from -0.46 to 0.21, with the minimum at QZ-NAMORS and the maximum at CN-HaM.

For daily GPP, the model also performs well in depicting the seasonal variation. However, in certain stations (e.g., DMCJZ) the peak GPP values within a year appear to be underestimated. In terms of the cropland flux sites, the GPP also shows double peaks within a year because of the double-cropping system (e.g., winter wheat and maize rotation), which is similar to ET. This is especially apparent for the GTZ, DXZ, and CF-YCA flux sites located in the North China Plain. For the site mean GPP, the discrepancies between the model and EC observations mainly range from -0.66 to 0.96 g C m$^{-2}$ d$^{-1}$ for most

flux sites except CF-YCA, MYZ, and DXZ, where the differences exceed 1.3 g C m$^{-2}$ d$^{-1}$.

### 3.2 Comparing with PML-V2 and other products

### 3.2.1 Comparisons at a plot-scale using EC observed data

To explore whether ET and GPP of PML-V2(China) simulations are more accurate than the previous products, we also evaluated ET and GPP accuracy from its global version, MOD16A2, MOD17A2H, and other five widely available ET or

GPP products against EC observations from the 26 sites at a daily or 8-day resolution. In this study, PML-V2(China) uses its cross-validated simulations to compare with other products instead of the calibration results to avoid introducing a priori knowledge. Additionally, to compare at a consistent time resolution, PML-V2(China) estimates in cross-validation mode





need to be up-scaled to an 8-day average or remain a daily scale, depending on the temporal resolution of the comparison products. Specifically, PML-V2(China) is compared to GLEAM and SEBAL at the daily scale, compared to PML-V2(Global), MOD16A2, MOD17A2H, EC-LUE, and VPM at the 8-day scale.

Table 3 provides a direct comparison of model performance among varieties of ET or GPP products against observations overall from 26 ground stations. It is evident that PML-V2(China) excels other state-of-the-art ET or GPP products, presented by *NSE* being 0.12 ~ 7.76 higher for ET and 0.07 ~ 0.79 higher for GPP, *R* being 0.07 ~ 0.68 higher for ET and 0.04 ~ 0.51 higher for GPP, and *RMSE* being 0.15 ~ 3.62 mm d$^{-1}$ lower for ET and 0.24 ~ 1.98 g C m$^{-2}$ d$^{-1}$ lower for GPP. Specifically, at a daily scale, PML-V2(China) ET exhibits the highest *NSE* value of 0.66, followed by GLEAM (0.44) and SEBAL (-7.10). PML-V2(China) daily ET achieves the highest *R* (0.84), followed by GLEAM (0.69) and SEBAL (0.16); correspondingly it obtains the smallest *RMSE* (0.33 mm d$^{-1}$), followed by GLEAM (1.04 mm d$^{-1}$) and SEBAL (3.94 mm d$^{-1}$). SEBAL is the worst performer, although its *Bias* of the closest to 0 (Table 3) because it is far away from the observed values significantly yielding a *Bias* value over 50% or less than -50% among five PFTs (Fig. 7). At the 8-day scale, PML-V2(China) outperforms PML-V2(Global) and MOD16A2 for estimating ET, with the highest value of *NSE* (0.74), *R* (0.87), and the lowest *RMSE* (0.66 mm d$^{-1}$). Moreover, PML-V2(China) has also the best performance in estimating 8-daily GPP, followed by PML-V2(Global), MOD17A2H, VPM, and EC-LUE, indicated by three statistics: *NSE*, *R,* and *RMSE*. In summary, PML-V2(China) performs well when compared with other mainstream ET or GPP products.

Figure 7 displays the performance comparison of four ET products with PML-V2(China) under nine PFTs. The simulated ET by PML-V2(China) has greater *NSE* and *R*, and less *RMSE* values than the other four products in most PFTs, especially in EBF, SAV and WET. All models have poor performance with *NSE* being lower than 0 except for PML-V2(China) in EBF and SAV. But PML-V2(China) is not the best in both ENF and BSV. In BSV, most models perform poorly, rendered by *NSE* being lower than 0 except for GLEAM (0.50) and PML-V2(China) (0.07 for the daily scale; 0.11 for the 8-day scale). And only SEBAL achieves worse results than PML-V2(China) in ENF. As shown in Fig. 8, PML-V2(China) performs significantly better than other advanced methods in CRO, MF, ENF, EBF, SAV, and BSV, producing higher *NSE*, *R,* and lower *RMSE* and *Bias*. While PML-V2(China) ranked second in GRA (following MOD17A2H), OSH (following PML-V2(Global)), and WET (following PML-V2(Global)). Synthetically, PML-V2(China) successfully captures the sites' seasonality in most PFTs compared to the high-resolution ET/GPP datasets currently available.

### 3.2.2 Comparisons at the basin-scale using ET$_{wb}$

In addition to testing the model at a plot-scale, Figure 9 (a-e) presents the ET validations from PML-V2(China) and four ET products based on RS against the annual ET$_{wb}$ in the 10 major river basins of China during 2003-2013. It illustrates that PML-V2(China) shows the best performance among them, as indicated by the highest *NSE* (0.82) and the lowest *RMSE* (69.59 mm yr$^{-1}$) and *Bias* (6.28%) values. This is closely followed by the GLEAM and PML-V2(global) with *NSE* values of 0.36 and 0.26, respectively. However, MOD16A2 and SEBAL tend to overestimate ET in the majority of basins with much smaller *NSE* values of -0.21 and 0.02, respectively, which are consistent with the performance evaluations using the EC



observation shown in Fig.7. Above basin-wide evaluations, together with the plot-scale validations against EC observation data, demonstrate PML-V2(China) overall performing best among the tested products selected in this study.

Fig. 9 (f) illustrates an inter-basin comparison of the 11-year mean ET of 2003–2013 within the five ET products. PML-V2(China) performs well in most basins (*Bias*es within ±15%) except for the Northwest and Southwest Basins, where ET is overestimated by 24.96% and 61.57%, respectively. Even so, PML-V2(China) performs best in the Southeast River Basin with *Bias* of -3.80%, which is still better than the five ET products selected herewith *Bias* ranging from 15.93% to 62.42%. Although PML-V2(China) overestimates ET in the Northwest to a large extent, it performs best relative to its global version (84.88%) and another ET product MOD16A2 (152.70%).

### 3.3 Spatial patterns and annual variations of ET, $E_c$, $E_i$, $E_s$, GPP, and WUE

Fig. 10 illustrates the spatial distribution of the multi-year (2001-2020) mean annual ET and three components (i.e., $E_c$, $E_i$, and $E_s$) from PML-V2(China) across China. In general, the ET shows an increasing gradient from the northwest to the southeast (Fig. 10a). High annual ET (> 900 mm) is mainly located in the water bodies, Hainan Island, and western Taiwan, while most parts of the Northwest River Basin exist the low annual ET (< 100 mm), especially in the western Inner Mongolia and Gansu, and southern Xinjiang. Annual ET experiences a statistically insignificant increasing trend during 2001-2020 with a tendency of 0.65 mm yr$^{-1}$ ($p > 0.05$). On the whole, the mean annual ET over China is 393.41 ± 10.9 mm yr$^{-1}$ (mean ± standard deviation) over the last 20 years. For three components, $E_c$ and $E_i$ products display a similar spatial distribution with annual ET, while high $E_s$ values (> 400 mm yr$^{-1}$) are mainly scattered in higher soil moisture content areas including the Tibet Plateau, Pearl River Delta, and Yangtze River Delta (Fig. 10b-d). High $E_c$ (> 600 mm yr$^{-1}$) and $E_i$ (> 80 mm yr$^{-1}$) values overall occur in the tropical and subtropical forests (e.g., Southwest River and Pearl River Basins), but low $E_c$ (< 50 mm yr$^{-1}$) and $E_i$ (< 5 mm yr$^{-1}$) values in the Northwest River Basin except for the Tianshan, Altai, and Qilian mountains. Especially, low $E_i$ values also appear in the cropland areas, such as the Northeast Plain, North China Plain, and Sichuan Basin (Fig. 10c). For annual variation over China, $E_c$ and $E_i$ increase significantly during 2001-2020 with a rate of 0.75 and 0.16 mm yr$^{-1}$ ($p < 0.001$), respectively. However, annual $E_s$ shows a declining trend with an insignificant rate of −0.27 mm yr$^{-1}$ (p < 0.05).

The mean annual GPP shows similar spatial patterns compared to the mean annual ET, as indicated by Fig. 10a and Fig. 11a. High annual GPP (> 2000 g C m$^{-2}$) mainly occurs in the tropical and subtropical forests and North China Plain where there exists the double-cropping system, but low annual GPP (< 100) in the arid zones such as the Northwest River Basin. On average, the multi-year GPP over China is 728.86 ± 53.9 g C m$^{-2}$, and interannual change displays a steady rise trend with a rate of 8.51 g C m$^{-2}$ yr$^{-1}$ ($p < 0.001$) since 2001. Using the coupled estimation of PML-V2(China) model, we study WUE (GPP divided by ET) during 2001-2020 across China (Fig. 11b). This result indicates the high annual WUE (>3 g C m$^{-2}$ H$_2$O) occurring in the forests and cropland, particularly in Northeast China and North China Plain. The annual variation of WUE is similar to that of GPP, with a significant increasing trend (Slope = 0.02, $p < 0.001$).



## 4 Discussion

### 4.1 Magnitude and trend in annual ET and GPP over China

For annual ET over China, the multi-year (2001-2020) mean annual ET from PML-V2(China) is 393.41 ± 10.9 mm (Fig. 10a). This result is overall consistent with previous studies (Yin et al., 2021; Cheng et al., 2021; Ma et al., 2019a), which also the country-wide averaged annual ET over China range from 359 ~ 482 mm yr$^{-1}$ based on a variety of models including the machine learning, RS, and land surface models. Furthermore, previous studies by Ren et al. (2015) and Wang et al. (2012a) show that the long-term mean precipitation and runoff in China are about 720 mm yr$^{-1}$ and 280 mm yr$^{-1}$, respectively.

Hence, it is believed that an annual ET less than 440 mm could be reasonable in China (Ma et al., 2019). The annual mean GPP over China from our results is 728.86 ± 53.9 g C m$^{-2}$ (Fig. 11a), which is lower than that of Jia et al. (2020) (771 g C m$^{-2}$), and higher than those of Yao et al. (2018) and Ma et al. (2019b) (690 g C m$^{-2}$ and 710 g C m$^{-2}$, respectively). These differences may associate with the distinctions in the time window and data sources (Jia et al., 2020).

The PML-V2(China) ET has a slightly increasing trend but with not statistically significant from 2001 to 2020, which is
consistent with the calculated ET using the Budyko equation (Feng et al., 2018; Su et al., 2022). In terms of annual GPP, we found that there is a significant ($p < 0.001$) increasing trend with a rate of 8.51 g C m$^{-2}$ yr$^{-1}$ during 2001-2020, in line with some other studies (Ma et al., 2019b; Yao et al., 2018; Ma et al., 2018). The most likely reason for the remarkable rise in GPP is the positive effect of ecological restoration projects in China (Tong et al., 2018). In fact, a large number of ecological restoration projects have been conducted since the 1990s, such as the Grain for Green project (Cao et al., 2009). These
findings also confirmed that a significant increase in vegetation growth occurred in China over the past years, which agreed well with Ma et al. (2018).

### 4.2 Advantages of this new dataset

The multi-scale testing using EC observations and water balance showed that accuracy in ET and GPP by the present PML-V2(China) is better than the global product of PML-V2 and other mainstream ET or GPP models (Table 3). The reasons may
be twofold. The first is that the water-carbon coupled process is particularly important for estimating ET and GPP since the water and carbon process are highly coupled by the stomatal aperture at the leaf level. This result was also supported by Xiao et al. (2013) and Zhang et al. (2019) in their recent studies. Second, this study employed 26 EC observations to calibrate the PML-V2 in China, which shows better accuracy than the previous global-scale ET and GPP estimates that were obtained using few EC observations to constrain the parameters. This indicates that more EC observations will facilitate the
improvement of ET and GPP estimates. In fact, although the EC sites of MOD16A2 (72 EC sites), GLEAM (91 EC sites), and PML-V2(Global) (95 EC sites) are more than in this study, there are only 0, 8, and 8 sites in China, respectively (Table 1). In particular, the SEBAL model only used eight EC sites for three PFTs (i.e., forests, cropland, and grassland).

In addition to the advantages of the overall accuracy in ET and GPP in the present study, the PML-V2(China) showed its strong ability to reveal the characteristics of the water consumption from the croplands that have the double-cropping system.



The GTZ in Hebei, DXZ in Beijing, and CF-YCA in Shandong are the only three observed sites with winter wheat - summer maize rotation cropping systems. We compared the intra-annual variation of the simulated ET and GPP between PML-V2(China) and other products against the EC-observed values at the three cropland sites (Fig. 12). In theory, when the winter wheat is harvested, the ET or GPP should decline to their valley values in June, which often occur between the two peaks (i.e., the reproductive growth stage of winter wheat and summer maize, respectively) within a given year. With this in mind, 

it can be seen that PML-V2(China) has improved its ability when compared to its global version, as has been indicated by its better performance in capturing the time when the lowest ET and GPP values emerged. This is mainly because an improved weighted Whittaker smoother was carried out to get better quality of LAI, as described in Section 2.2.1. While the GLEAM is also able to detect the time when the valley values appeared, it underestimates ET evidently during the wheat growing season. In terms of the SEBAL and MOD16A2, both have much poorer performances in detecting such intra-annual 

variations in ET. Regarding other GPP models, only MOD17A2H can catch the time when the valley values appear. However, it substantially underestimates GPP in winter both wheat and summer maize growing seasons.

### 4.3 Implications of PML-V2(China)

Based on the substantial advantages discussed above, PML-V2(China) has great implications and application prospects. For instance, daily outputs from PML-V2(China) can be better used by the agricultural and water sectors for operational 

applications. Timely access to daily data at the regional or national scale helps the Ministry of Agriculture and Water Resources to develop better policies. Indeed, there is a remarkable relationship between soil water content and ET (Graf et al., 2014; Brust et al., 2021), so getting daily ET information accurately is of great significance for soil water depletion assessment, irrigation system design, and water resources management in agricultural areas, such as in the North China Plain. On the other hand, this dataset has better simulations of carbon consequences and water use efficiency, which is important 

for carbon-neutron policy (Yang et al., 2022). According to this study, vegetation in China exhibits a huge potential for carbon sequestration in the water-carbon cycles and plays an important role in the global carbon cycle.

### 4.4 Uncertainties

### 4.4.1 Eddy-covariance method and water-balance formula

Although PML-V2(China) showed relatively good performance when compared to the EC sites and water-balance-based 

evapotranspiration ($ET_{wb}$), there exist several uncertainties related to the observed data (i.e., flux sites and $ET_{wb}$). First, the EC technique, considered as a standard way to measure surface fluxes (Aubinet et al., 1999; Liu et al., 2011; Baldocchi, 2014), meets some issues as well including corrections when processing the turbulence data and from energy non-closure problem. The corrections, such as the spike detection, lag correction of $H_2O$ and $CO_2$ based on the vertical wind, coordinating rotation, corrections for density fluctuation, and frequency response correction, had been pre-processed before 

the investigators shared data. Nevertheless, there is evidence that diverse data processing designs may lead to errors of 10%



~ 15% (Mauder et al., 2007). Besides, systematic bias in device, the loss from the contribution of low-frequency eddies to energy transmission and the ability to capture larger eddies and the secondary circulations, could cause the energy-imbalance problem (Liu et al., 2011). Hence, there are usually two schemes to deal with the energy non-closure issue for EC users, that is, to perform energy closure correction (Cheng et al., 2021), or to maintain the original four-component data including latent
heat (LE), sensible heat (H), soil heat flux (G) and net radiation ($R_n$) (Zhang et al., 2019; Ma and Zhang, 2022). We chose the second method in this study, considering that (i) forcing energy closure will introduce new errors artificially, and (ii) most Chinese EC observation towers lack G and $R_n$. The observed ET calculated from latent heat flux of the site without energy closure correction will be slightly less than the real value, resulting in a smaller ET simulated by the model determined by calibration using the sites. This phenomenon is not fully reflected in comparing with the basin ET based on
the water-balance calculation that only in the Pearl and Southeast River Basin PML-V2(China) underestimates multi-year mean ET (Fig. 9f), because the $ET_{wb}$ used for assessing the simulation performance of models also needs to be explored its accuracy.

Second, the inconsistency of the grid cell and EC footprint could also result in uncertainty when compared rudely to the measurements. Generally, the EC towers have a footprint of 100 -1000 $m^2$, which is usually decided by tower height and
heterogeneity of the underlying surface (Liu et al., 2016; Xu et al., 2017). For example, the footprint of the forest sites is larger than the grassland and wetland sites (Chen et al., 2012). In this study, PML-V2(China) model is first calibrated at 500m grid cell avoiding the inconsistency issue to some extent. However, there still exists a mismatch between the grid cell center and EC sites. In this regard, higher spatial resolution products or additional ground observations at relevant scales would be beneficial for the cross-validation of the modelling grid cell (Ma et al., 2019a). Note also that the limited flux sites
for some PFTs may introduce extra uncertainties for model parameters since only one site was available for the OSH, SAV, EBF, and MF in this study. In fact, we artificially construct multiple site samples in the above PFTs sites, utilizing the characteristics of the long time series of these sites. Nevertheless, the terrestrial biome of the sole flux site maybe not typical in other climate zones for the same PFT (Cheng et al., 2021). Therefore, more flux sites for the same PFT are necessary to calibrate the model.

Third, ET derived from water-balance over China was invested as a reference at the basin-scale, although the results may be affected by some sources of uncertainty. For instance, the applicability of water-balance relies on its formula composition. In this paper, the water storage change (in Equation 7) from GRACE was included for purpose of reducing the uncertainties in estimating annual $ET_{wb}$. Those studies not using TWSC in $ET_{wb}$ may not explain the decreasing groundwater such as in the Hai River Basin due to the human water extraction, which is not conducive to the credibility of the verification results
(Cheng et al., 2021). In addition, the precision of its input data also affects the reliability of $ET_{wb}$ (Mao and Wang, 2017). Precipitation, as the main source for ET, impacts $ET_{wb}$ to varying extent. Nevertheless, precipitation data are derived from observations of field rain gauge network and its usability relies on the intensive and high-quality ground observations, which makes the Prcp estimates from statistics of stations worse in the less populated remote regions or areas having highly various topography, particularly in the west China (Immerzeel et al., 2015; Tang et al., 2016; Zhong et al., 2019). Zhong et al. (2019)





evaluated three precipitation products in China and found that there was a slight overestimation in the west of China and an obvious underestimation in the west-Tibet Plateau. Accordingly, it could be stated that $ET_{wb}$ was overestimated in Pearl and the Southeast and underestimated in the Southwest River basin using Eq. 7.

### 4.4.2 Input data

While the daily ET and GPP of the PML-V2(China) product (in the calibration mode) simulated well against 26 flux sites
overall (Fig. 3) and in most PFTs (Fig. 4), PML-V2(China) in calibration is degraded compared to its cross-validation (Fig. 4), such as in GRA. In this study, we got one parameter set for the GRA type by employing eight sites, including the sparse grassland (QZ-BJ) and the dense grassland (QZ-NAMORS). Although it may be appropriate to use diverse parameter values for estimating ET and GPP by further dividing the grassland type into finer land types, this comes at the expense of ignoring the possible interannual changes in land types because few LUCC maps with fine classifications and annual resolutions
simultaneously (Ma and Zhang, 2022).

PML-V2(China) mainly used the remote sensing and meteorological data (e.g., MODIS, CMFD, GLDAS-2.1, and ERA5) as the inputs (see section 2.2). However, there are still some uncertainties in these data (Zhang et al., 2019; Cheng et al., 2021). For example, we used the land cover datasets (MCD12Q1.006) as the PFTs data across China. However, there exist misclassification issues for MCD12Q1 because of spectral confusion (e.g., savannas and grasslands) and coarse resolution
(e.g., the mixed pixel of cropland and natural vegetation) (Zhang et al., 2019; Liang et al., 2015; Adzhar et al., 2022). Moreover, LAI is a critical variable describing vegetation growth, and its temporal changes affect stomatal conductance and further affect the transpiration rate from the vegetation canopy. Using LAI, PML-V2(China) model simulated $E_c$ and $E_i$ components at the canopy scale. To avoid noise issues caused by clouds, shadows, snow, and so on, MODIS LAI in this study has been smoothed by the weighted Whittaker smoother which can deal with underestimation and inefficiency issues
(Kong et al., 2019). However, there are still underestimates in the sparse vegetation areas. This may explain why the ET and GPP estimates are poor in BSV (Fig. 7 and Fig. 8).

To extend the simulation period, we used GLDAS-2.1 meteorological forcing data during 2019-2020 since the CMFD dataset is only up to 2018. To check if using these two datasets generates a systematic bias, we reran the PML-V2(China) in 2001-2018 using GLDAS-2.1 and compared the modelling results with those obtained using CMFD (Fig. S1). At the
national scale, the mean *difference*, calculated by $(\text{PML-V2(China)}_{GLDAS\text{-}2.1} - \text{PML-V2(China)}_{CMFD})/\text{PML-V2(China)}_{CMFD}$, varied from -1.22% to 1.62% among $E_i$, $E_c$ and GPP, and was 13.72% for $E_s$ and 7.78% for ET. The *difference* is within -25% ~ 25% in more than 66% of the research region for all five variables (Fig. S1b2-e2), specifically 100% for GPP, 95% for $E_c$, 84% for ET, 73% for $E_i$, and 66% for $E_s$ (Fig. S1b3-e3). This illustrates that PML-V2(China) using the GLDAS-2.1 in 2019-2020 does not generate a noticeable systematic deviation.

Additionally, downscaling uncertainties could be also introduced by the bilinear interpolation method which has been applied to minimize the footprint impact of coarse meteorological inputs, such as CMFD (Fig. 1). This approach depends only on nearby grid cells to downscale, which could neglect the other relative supports. For instance, precipitation is not only

related to the surrounding precipitation but also location and terrain (e.g., elevation and aspect) (Yue et al., 2020). Chao et al. (2018) found that gridded precipitation products in the high-altitude regions are far below what is inversely inferred by
glacier mass balances. Consequently, the geographically weighted regression method coupled with a weighting function could work well to interpolate meteorological data (Chao et al., 2018).

## 5 Conclusions

This study developed a daily, 500m ET and GPP data product (PML-V2(China)) using the locally calibrated water-carbon coupled model, PML-V2. The model has been well-calibrated against observations at 26 flux sites across nine plant
functional types, and it performs satisfactorily in the cross-validation mode. More importantly, the plot- and basin-scale evaluations suggest that the newly developed product outperforms not only the global version of PML-V2 but also other mainstream RS-based ET and/or GPP products. With such a new product, we investigated spatial patterns and trends in ET and its components ($E_c$, $E_i$, $E_s$), GPP, and WUE from 2001-to 2020 across China. In short, the present PML-V2(China) product has the following advantages: (i) the water output is tightly constrained by carbon flux; (ii) it has high spatial and
temporal resolutions simultaneously; (iii) it obtains the improved accuracy in ET and GPP across different plant functional types because of the optimal parameter sets for China by exploiting 26 EC sites; and (iv) showing the better ability to reveal the ET and GPP for the croplands with the double-cropping system. In summary, we provide a novel daily and 500m resolution ET and GPP product across China, which can be used by research communities and various water and ecological departments for operational applications.

## Data availability statement

The product named PML-V2(China) with daily and 500m resolutions from February 26, 2000 to December 31, 2020 is freely available at the National Tibetan Plateau Data Center (http://dx.doi.org/10.11888/Terre.tpdc.272389, Zhang and He, 2022).

## Author contributions

SH and YZ designed the research. SH collected the EC observations and other input data. DK provided the improved LAI data. SH wrote the paper. YZ, NM, JT, DK and CL revised the paper.

## Competing interests

The authors declared that they have no conflict of interest.



**Acknowledgements**

We gratefully acknowledge all people participating in EC measurements and uploading the observed data to available platforms (i.e., the National Tibetan Plateau Data Center, the Heihe Integrated Observatory Network, theChinaFLUX, and FLUXNET2015). We are grateful to the Google Earth Engine team for the data distribution and online computing. We thank NOAA for providing $CO_2$ concentration at ftp://aftp.cmdl.noaa.gov/products/trends/co2/co2_mm_gl.txt.

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



**Table 1: Summary of typical ET and GPP products with high temporal and/or high spatial resolutions.**

| Variable | Dataset abbreviation | Spatial resolution | Temporal resolution | Temporal coverage | Principle or model | EC evaluation | Reference |
|---|---|---|---|---|---|---|---|
| ET | MOD16A2 | 500m | 8-day | 2001-present | PM | 72 EC sites in AmeriFlux (no using sites in China) | Mu et al. (2011) |
| ET | SEBAL | 1000m | daily | 2001-2018 | one-source model of surface energy balance residual | 8 EC sites in China | Cheng et al. (2021) |
| ET | GLEAM | 0.25° | daily | 1980-2020 | P-T | 91 global EC sites (including 8 sites in China) | Miralles et al. (2011a) and Martens et al. (2017) |
| GPP | MOD17A2H | 500m | 8-day | 2000-present | LUE | Not performed. | Running et al. (2015) |
| GPP | VPM | 500m | 8-day | 2000-2019 | LUE | 113 global EC sites (including 8 sites in China) | Zhang et al. (2017), Zhang et al. (2017); Zhang et al. (2021) |
| GPP | EC-LUE | 0.05° | 8-day | 1982-2018 | LUE | 95 global EC sites (including 7 sites in China) | Zheng et al. (2020) |
| ET, GPP | PML-V2 (Global) | 500m | 8-day | 2000-2020 | PML-V2 | 95 global EC sites (including 8 sites in China) | Zhang et al. (2019) |
| ET, GPP | PML-V2 (China) | 500m | daily | 2000-2020 | PML-V2 | 26 EC sites in China | This study |



**Figure 1: Flowchart of EC flux and AWSs data pre-processing and PML-V2 model processing which is used to convert RS images and meteorological forcing images into GPP, E$_c$, E$_i$, E$_s$, E$_w$, and ET. For pre-processing part: NEE (Net Ecosystem Exchange, µmol m$^{-2}$ s$^{-1}$), LE (Latent heat, W m$^{-2}$), Rg (incoming radiation, W m$^{-2}$), rH (relative humidity, %), Ta (air temperature, °C), and QC (Quality Control). For the PML-V2 model part: Tmax (daily maximum temperature, °C), Tmin (daily minimum temperature, °C), Tavg (daily mean temperature, °C), Pa (atmosphere pressure, kPa), U (wind speed at 10-m height, m s$^{-1}$), q (specific humidity, kg kg$^{-1}$), Prcp (precipitation, mm d$^{-1}$), Rl (inward longwave solar radiation, W m$^{-2}$), Rs (inward shortwave solar radiation, W m$^{-2}$), Pi (the difference of Prcp and Ei, mm d$^{-1}$), Es_eq (equilibrium evaporation, mm d$^{-1}$), ET_w (evaporation from water body, snow and ice, mm d$^{-1}$) and GEE (Google Earth Engine).**





**Figure 2: Geographical locations of 26 EC flux towers for nine major IGBP PFTs, the main rivers and the ten major river basins in China. Overlain are 20-year mean annual aridity index (AI) values during 2001-2020 using GLDAS-2.1, that is, the ratio of annual precipitation to Penman potential evapotranspiration. PFTs shown in legend are ENF (Evergreen Needleleaf Forests), EBF (Evergreen Broadleaf Forests), MF (Mixed Forests), OSH (Open Shrublands), SAV (Savannas), GRA (Grasslands), WET (Permanent Wetlands), CRO (Croplands), and BSV (Barren Sparse Vegetation).**




**Table 2: Details of 26 EC flux towers employed in this study. Note that AP indicates mean annual precipitation and AT refers to mean annual temperature in its observed period.**

| Site code | Site name | IGBP | Latitude (°E) | Longitude (°N) | AP (mm yr-1) | AT (°C) | Time coverage | References |
|---|---|---|---|---|---|---|---|---|
| ARCJZ | Arou | GRA | 38.0473 | 100.4643 | 521 | -2.7 | 2013-2017 | Liu et al. (2018) |
| BNXJL | Xishuangbanna rubber | EBF | 21.9000 | 101.2667 | 1765 | 22.1 | 2013 | Yu et al. (2021) |
| CF-CBF | Chinaflux Changbai forest | MF | 42.4025 | 128.0958 | 608 | 4.3 | 2003-2010 | Zhang et al. (2006a) |
| CF-HBG_S01 | Chinaflux Haibei grassland | OSH | 37.6653 | 101.3311 | 610 | -5.9 | 2003-2010 | Hui et al. (2021) |
| CF-HBG_W01 | Chinaflux Haibei wetland | WET | 37.6086 | 101.3269 | 616 | -3.9 | 2004-2006 | Zhang et al. (2020a) |
| CF-NMG | Chinaflux Neimengu grassland | GRA | 43.3233 | 116.4036 | 387 | 1.2 | 2004 | Hao et al. (2020) |
| CF-QYF | Chinaflux Qianyanzhou forest | ENF | 26.7414 | 115.0581 | 1490 | 19.3 | 2004-2006 | Wen et al. (2006) |
| CF-YCA | Chinaflux Yucheng | CRO | 36.8290 | 116.5702 | 602 | 14.8 | 2006-2007 | Zhao et al. (2021) |
| CN-Cng | Changling | GRA | 44.5934 | 123.5092 | 364 | 6.5 | 2007-2010 | Dong et al. (2011) |
| CN-Du2 | Duolun_grassland (D01) | GRA | 42.0467 | 116.2836 | 388 | 3.0 | 2006-2008 | Chen et al. (2009) |
| CN-HaM | Haibei Alpine Tibet site | GRA | 37.6975 | 101.2733 | 534 | -4.0 | 2002-2004 | Kato et al. (2006) |
| DMCJZ | Daman | CRO | 38.8555 | 100.3722 | 163 | 9.2 | 2017 | Liu et al. (2018) |
| DSLZ | Dashalong | WET | 38.8399 | 98.9406 | 346 | -8.3 | 2015-2018 | Liu et al. (2018) |
| DXZ | Daxing | CRO | 39.6213 | 116.4271 | 547 | 12.7 | 2010 | Liu et al. (2013) |
| DYKGTSLZ | Dayekouguantan forest | ENF | 38.5337 | 100.2502 | 228 | 0.2 | 2010-2011 | Li et al. (2009) |
| GTZ | Guantao | CRO | 36.5150 | 115.1274 | 433 | 14.0 | 2008 | Liu et al. (2013) |
| HLZ | Huailai | CRO | 40.3491 | 115.7880 | 377 | 10.2 | 2014 | Liu et al. (2013) |
| HZZHMZ | Huazhaizi Desert Steppe | BSV | 38.7659 | 100.3201 | 167 | 8.7 | 2017 | Liu et al. (2018) |
| MYZ | Miyun | CRO | 40.6308 | 117.3233 | 584 | 9.0 | 2008 | Liu et al. (2013) |
| QZ-BJ | Tibetan Plateau BJ | GRA | 31.3688 | 91.8988 | 460 | 0.2 | 2011-2013 | Ma et al. (2020) |





| QZ-NAMORS | Tibetan Plateau NAMORS | GRA | 30.7730 | 90.9632 | 405 | -0.3 | 2008-2009 | Ma et al. (2020) |
|---|---|---|---|---|---|---|---|---|
| QZ-QOMS | Tibetan Plateau QOMS | BSV | 28.3607 | 86.9491 | 199 | 1.2 | 2015 | Ma et al. (2020) |
| YJGRHG | Yuanjiang dry-hot valley | SAV | 101.2667 | 21.9000 | 876 | 20.2 | 2014 | Yang et al. (2021) |
| YKGQLZZ | Yingke | CRO | 38.8569 | 100.4103 | 85 | 8.3 | 2011 | Liu et al. (2018) |
| YKZ | Yakou | GRA | 38.0142 | 100.2421 | 484 | -1.2 | 2016-2018 | Liu et al. (2018) |
| ZYSDZ | Zhangye wetland | WET | 38.9751 | 100.4464 | 146 | 8.8 | 2013-2018 | Liu et al. (2018) |



**Figure 3:** Scatter plots between the observed ET and GPP against PML-V2(China) simulations in calibration and cross-validation modes: daily comparisons in the left panels and site mean comparison in the right panels.





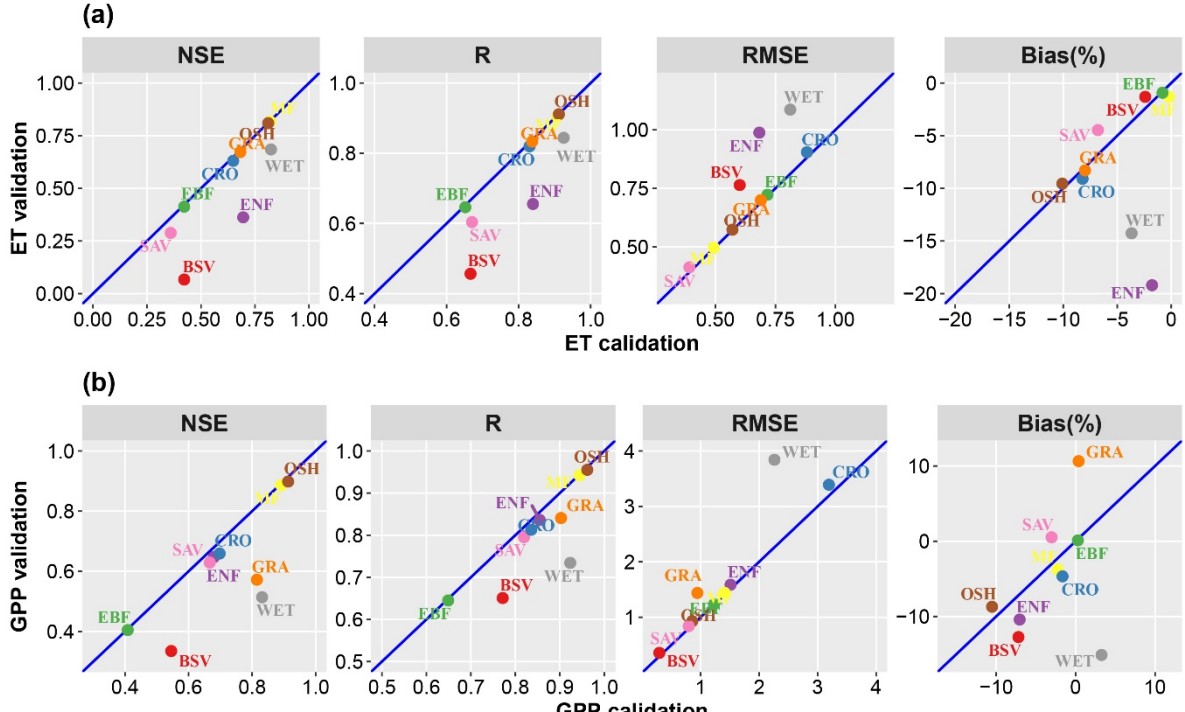

**Figure 4: Comparison of ET and GPP between PML-V2(China) model calibration and validation across ten PFTs.**




**Figure 5: The daily ET simulated by PML-V2(China) in calibration mode and the observed daily ET variation in time series from 26 EC sites (see Figure 2) across China. 'ALL' represents the site mean value for each EC site.**





**Figure 6: The daily GPP simulated by PML-V2(China) in calibration mode and the observed daily GPP variation in time series from 26 EC (see Figure 2) sites over China. 'ALL' represents the site mean value for each EC site.**





**Table 3: Statistical indicators of PML-V2(China) and other models for simulating ET and GPP at 26 EC flux towers. *NSE* and *R* values are unitless. The unit of *RMSE* for ET is mm d$^{-1}$ while it is g C m$^{-2}$ d$^{-1}$ for GPP. The unit of *Bias* is %.**

| Scale | Variable | Models | *NSE* | *R* | *RMSE* | *Bias* |
|---|---|---|---|---|---|---|
| daily | ET | PML-V2(China) | 0.66 | 0.84 | 0.33 | -7.97 |
| | | GLEAM | 0.44 | 0.69 | 1.04 | -14.45 |
| | | SEBAL | -7.10 | 0.16 | 3.95 | 5.31 |
| 8-day | ET | PML-V2(China) | 0.74 | 0.87 | 0.66 | -11.54 |
| | | PML-V2(Global) | 0.62 | 0.80 | 0.81 | -5.05 |
| | | MOD16A2 | 0.37 | 0.63 | 1.07 | -10.90 |
| 8-day | GPP | PML-V2(China) | 0.75 | 0.87 | 1.93 | -6.51 |
| | | PML-V2(Global) | 0.68 | 0.82 | 2.17 | -1.74 |
| | | MOD17A2H | 0.49 | 0.78 | 2.74 | -38.79 |
| | | EC-LUE | -0.04 | 0.35 | 3.91 | -41.91 |
| | | VPM | 0.21 | 0.60 | 3.41 | -8.21 |


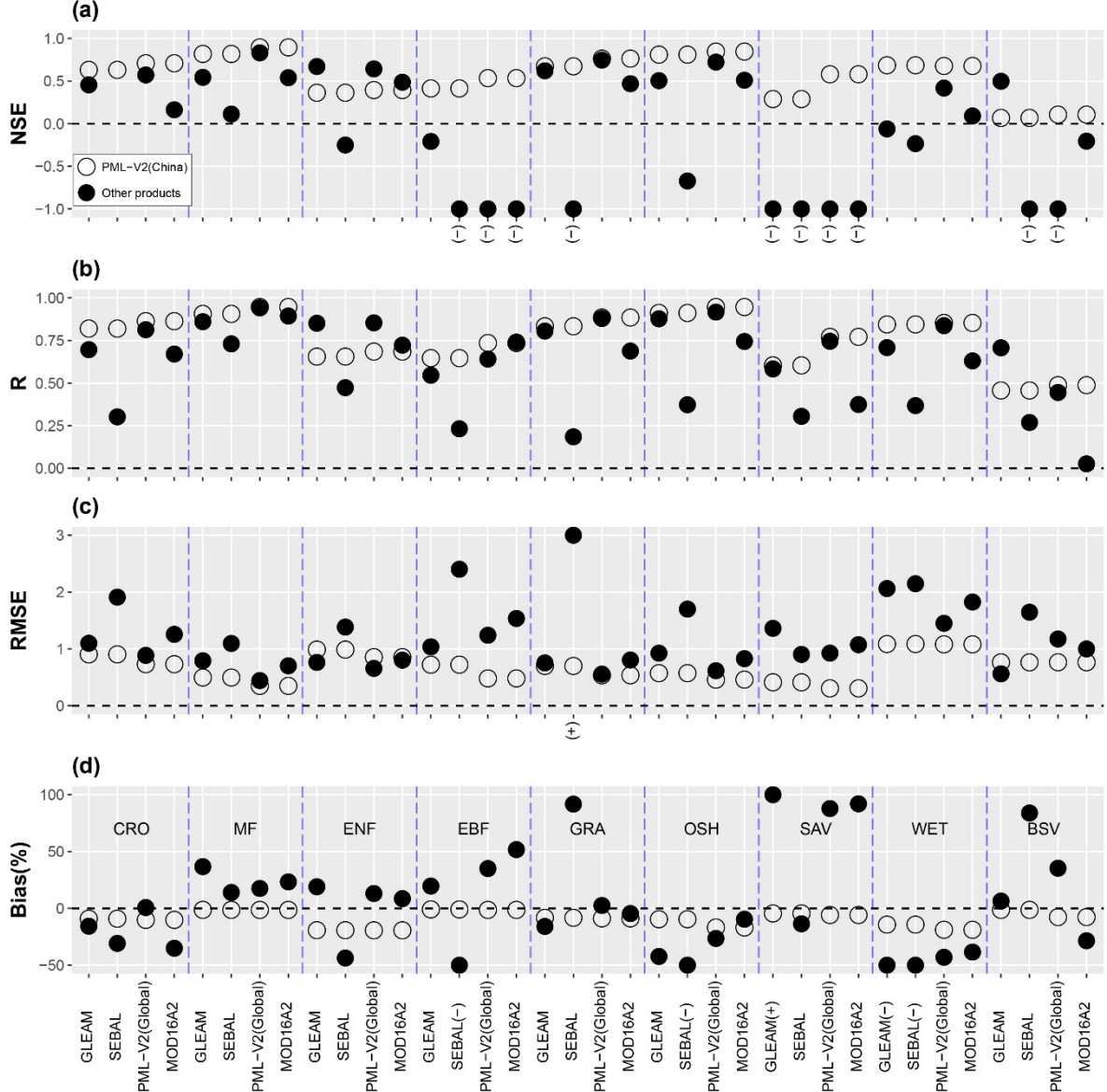

**Figure 7: Statistical indicators of PML-V2(China) and other models for estimating ET at each PFTs. Open and solid dots represent PML-V2(China) estimated ET in cross-validation mode and other models. For the daily temporal resolution of GLEAM and SEBAL, PML-V2(China) is also daily scale; while for 8-day resolution of PML-V2(Global) and MOD16A2, the referred PML-V2(China) is upscaled to 8-day. Note that '(+)' indicates the model's simulation statistics dot is more than the upper bound while '(-)' indicates the model's simulation statistics dot is less than the low bound.**

**Figure 8: Statistical indicators of PML-V2(China) and other models for estimating GPP at each PFTs. Open and solid dots**
**875 represent PML-V2(China) estimated GPP in cross-validation mode and other models. PML-V2(China) is upscaled to 8-day to**
**compare with the 8-day resolution of PML-V2(Global), MOD17A2H, EC-LUE, and VPM. Note that '(+)' indicates the model's**
**simulation statistics dot is more than the upper bound while '(-)' indicates the model's simulation statistics dot is less than the low**
**bound.**




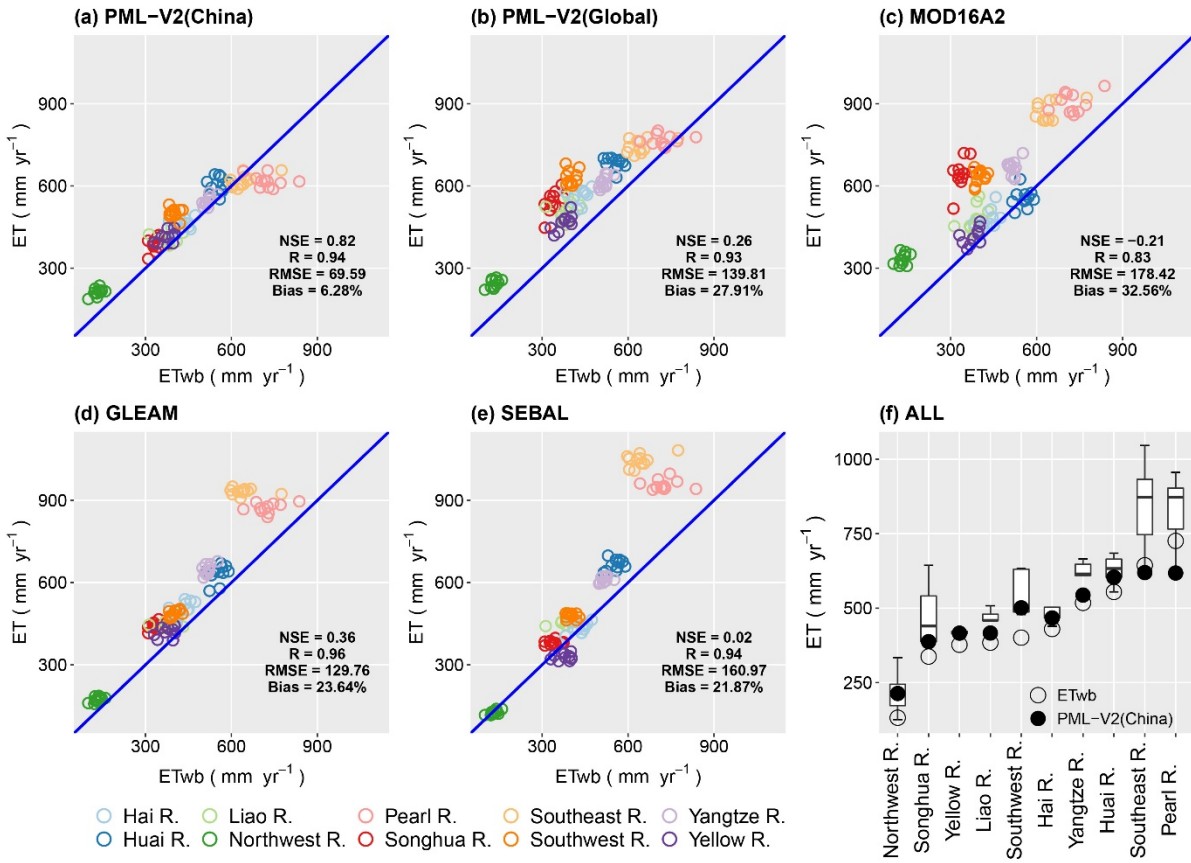

**Figure 9: Annual evapotranspiration (ET) of (a) PML-V2(China), (b) PML-V2(Global), (c) MOD16A2, (d) GLEAM, and (e) SEBAL plotted against the water-balanced derived ET (ET$_{wb}$) values for ten major river basins over China during 2003–2013. The boxplot in (f) shows a multi-year mean of the five ET products above per river basin.**



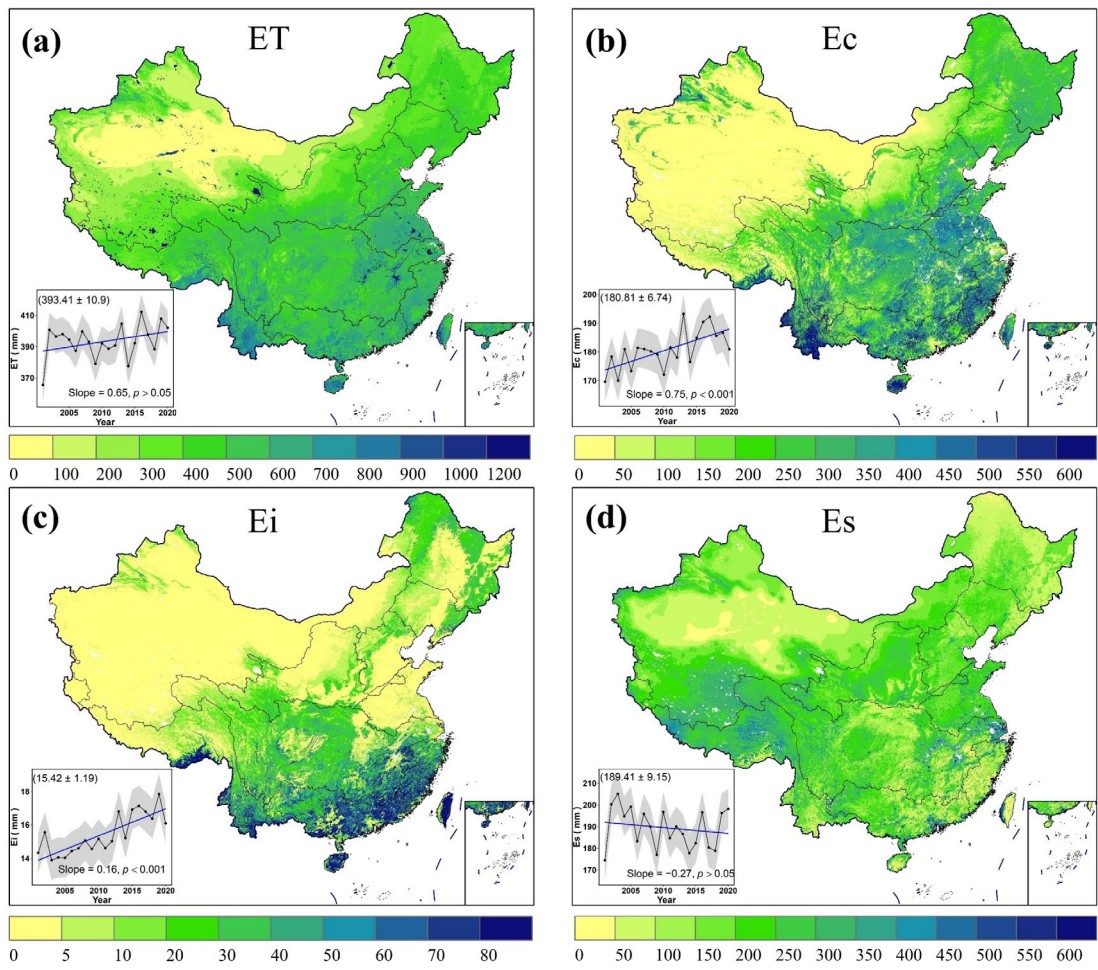


**Figure 10: Spatial pattern of mean annual ET, E$_c$, E$_i$, E$_s$ and their interannual changes during 2001–2020. In all insets, the shaded area represents the standard deviation of the simulated data. The number in the parentheses of each inset is mean ± standard deviation in the past 20 years over China.**




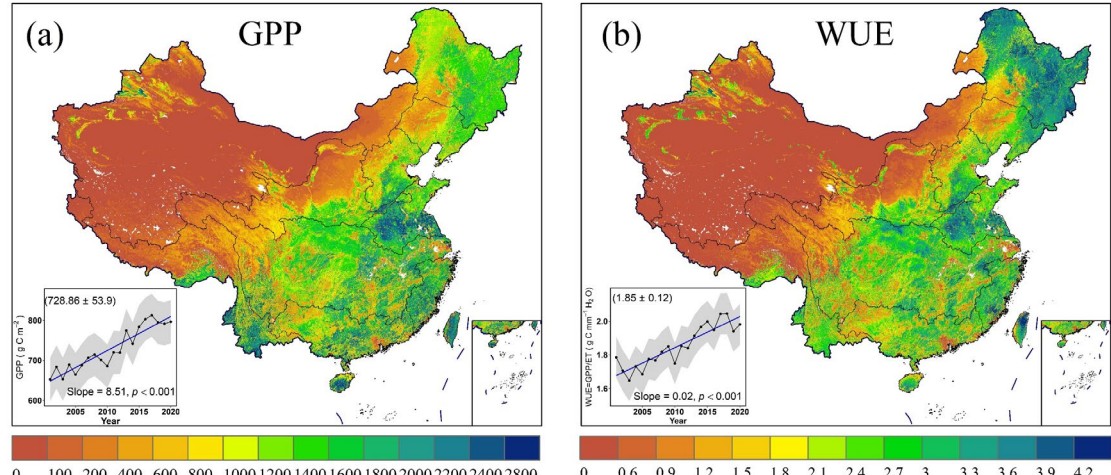

**Figure 11: Spatial pattern of mean annual GPP and WUE and their interannual changes during 2001–2020. The shaded area in the inset represents the standard deviation of the simulated data. The number in the parentheses of each inset is mean ± standard deviation in the past 20 years over China.**



**Figure 12: The intra-annual variation of (a) ET at three crop-rotation stations between the observed and the simulated by PML-V2(China) in validation mode, PML-V2(Global), SEBAL, GLEAM, and MOD16A2, respectively; (b) GPP at three crop-rotation stations between the observed and the simulated by PML-V2(China) in validation mode, PML-V2(Global), EC-LUE, VPM, and MOD17A2H, respectively. The blue dotted lines above pass through the lowest values between the two peaks of the observed ET or GPP per year. Note that all variables are averages every 8 days although units are per day.**