# Peer review of "A daily and 500m coupled evapotranspiration and gross primary production product across China during 2000-2020"

_Earth System Science Data, 2022_

## Author Comment (AC2)

**Reply to Reviewer 2**

**General Comments:**

He et al. constructed daily and 500m ET and GPP datasets in China using PML-V2. Compared with previous products, this model outputs improved in several aspects, including 26 EC sites being used for model calibration and validation, country-specific meteorological forcing, daily data, and intra-annual dynamics for multiple ecosystems. This ambitious work provides valuable data products for assessing the carbon and water cycles in China. They may also provide guidance in agricultural production and ecosystem management. The authors may consider the following suggestions to improve the robustness of this manuscript.

**Response:**

Thank you for appreciating our work and considering that the products are very valuable. We have carefully checked and re-edited the original manuscript. In the following, we reply to all comments in a point-by-point response. All comments are shown in blue. Sentences from the manuscript are in italics and the revised contents are indicated in red.

**Specific Comments:**

1. Line 99, the whole name for CMFD should be provided when it is first mentioned in the text.

**Response:**

We have added the whole name - the China Meteorological Forcing Dataset for the CMFD dataset.

2. Line 144-146, this sentence is not appropriate. You may use the MODIS land cover product, but it is debatable if it has the highest accuracy in China since there are many recently released land use/covered datasets with a high spatial resolution (30m and 10m). Many MODIS products based on the MODIS land use dataset may have low credibility in regions with complex terrain such as in the Loess Plateau.

**Response:**

We revised the sentence as follows:

*Here we used the International Geosphere-Biosphere Program (IGBP) layer of MCD12Q1.006 land cover product (Sulla-Menashe et al., 2019) during 2000-2020 since IGBP classification is annually continuous and has* red-*acceptable* accuracy in China when compared with *other* land cover products (Feng and Bai, 2019).*

3. Did you test the continuity between GLDAS-2.1 and CMFD?

**Response:**

Currently, we compared the magnitude and variability of the products using different meteorological forcing inputs, i.e., PML-V2(China)$_{GLDAS-2.1}$ and PML-V2(China)$_{CMFD}$, at the grid and national scale in section 4.4.2, as follows:

*To extend the simulation period, we used GLDAS-2.1 meteorological forcing data during 2019-2020 since the CMFD dataset is only up to 2018. To check if using these two datasets*

*generates a systematic bias, we reran the PML-V2(China) in 2001-2018 using GLDAS-2.1 and compared the modelling results with those obtained using CMFD (Fig. S1). At the national scale, the mean difference, calculated by $(PML\text{-}V2(China)_{GLDAS\text{-}2.1} - PML\text{-}V2(China)_{CMFD})/PML\text{-}V2(China)_{CMFD}$, varied from -1.22% to 1.62% among $E_i$, $E_c$, and GPP, and was 13.72% for $E_s$ and 7.78% for ET. The difference is within -25% ~ 25% in more than 66% of the research region for all five variables (Fig. S1b2-e2), specifically 100% for GPP, 95% for $E_c$, 84% for ET, 73% for $E_i$, and 66% for $E_s$ (Fig. S1b3-e3). This illustrates that PML-V2(China) using the GLDAS-2.1 in 2019-2020 does not generate a noticeable systematic deviation.*

The PML-V2(China) product of 2019-2020 is the interim data as the supplement of PML-V2(China) after 2018. We suggest that users do spatial variability analysis instead of trend analysis if they want to use the PML-V2(China) of 2019-2020. With the release of the meteorological dataset, we will continue to update the PML-V2(China) using the CMFD inputs. Moreover, we have removed the description and figures about the trend analysis of PML-V2(China) for 2019-2020 in the manuscript.

[Figure]

*Figure S1: The modelling results using GLDAS-2.1 meteorological forcing data during 2001-2018 and comparison with the PML-V2(China) product using CMFD: (a1-e1) Spatial distribution of the 18-year mean of five variables; (a2-e2) Spatial distribution of the difference using two forcing datasets, calculated by $(PML\text{-}V2(China)_{GLDAS\text{-}2.1} - PML\text{-}V2(China)_{CMFD})/PML\text{-}V2(China)_{CMFD}$; and (a3-e3) Proportion of difference in each river basin. 'ALL' represents the whole study area. The legends for (a3-e3) are the same as*

*that for (a2-e2). Taking Fig.(a3) as an example, the area percentage of ET difference in 0 ~ 25% in the Songhua River Basin is about 99%.*

**References**

He, J., Yang, K., Tang, W., Lu, H., Qin, J., Chen, Y., and Li, X.: The first high-resolution meteorological forcing dataset for land process studies over China, Sci Data, 7, 25, https://doi.org/10.1038/s41597-020-0369-y, 2020.

4. Section 2.6, the simulated model outputs were validated at the EC site-level, and compared with the publicly available dataset. How did you get the parameter set for a certain land use type? Did all the land use types have a unique parameter set? Did you run the model at each site?

**Response:**

(1) For each land use type, we used a global optimization method - genetic algorithm to gain the optimal solution by setting population size 1000 and number of generations 50 by minimizing an objective function including $ET_{obs}$ and $GPP_{obs}$. (2) Each of the nine land use types has a unique parameter set, so they are nine parameter sets. (3) Yes, we run the model at each site. We revise sections 2.5 and 2.6 to make the model calibration and model validation parts clearer, as follows:

*2.5 Model calibration and model validation*

*The 11 parameters of the PML-V2 model for each PFT were calibrated and cross-validated against 26 EC sites by a global optimization method - genetic algorithm (GA). The GA generates a randomly initialized population and then evaluates the fitness of solutions according to its objective function. As generations iterate, the population includes more appropriate solutions, and eventually, it will converge (Holland, 1992; Konak et al., 2006). Specifically, we applied the GA algorithm with population size 1000 and number of generations 50. All EC-observed ET and GPP data within a PFT are used to minimize the following objective function ($F_{opt}$):*

$$F_{opt} = 2 - NSE_{ET} - NSE_{GPP} = \frac{\sum_{i=1}^{N}(ET_{est}-ET_{obs})^2}{\sum_{i=1}^{N}\left(ET_{obs}-\overline{ET_{obs}}\right)^2} + \frac{\sum_{i=1}^{N}(GPP_{est}-GPP_{obs})^2}{\sum_{i=1}^{N}\left(GPP_{obs}-\overline{GPP_{obs}}\right)^2} \quad (8)$$

*where $NSE_{ET}$ and $NSE_{GPP}$ are the Nash-Sutcliffe Efficiency of the daily ET and the daily GPP, respectively. The subscripts est and obs stand for the estimated and the observed, respectively. In this way, each of the nine PFTs gained a unique set with 11 calibrated parameter values, illustrated in Table S1*

*The 'leave-one-out' cross-validation method was utilized to evaluate the robustness of the PML-V2 model (Zhang et al., 2019). For each PFT, the data from one "ungauged" observation was excluded from the optimization while the data from all other observations at the same PFT were used for model calibration to obtain the simulated at the "ungauged" position. All nine PFTs were actualized in this way. Note that the PFT including EBF, MF, OSH, and SAV only has one ground site (Table 2). Therefore, it is appropriate to divide the data in each of the four sites into two sub-groups for cross-validation. The CF-CBF and the CF-HBG_S01 covering from 2003 to 2010, were divided into two sub-groups, each of which had 4 years: 2003-2006 and 2007-2010. While both the BNXJL and YJGRHG only covered*

*one year and were divided into two sub-groups by a two-day time step, separately. After that, the daily estimates in the cross-validation mode were against the daily observation from the 26 stations to explore the model transferability from known observations to any location.*

**2.6 Model performance metrics**

*We assessed the performance of calibration and cross-validation of PML-V2 (and other seven mainstream ET and GPP products) against the observed sites or water-balance basins utilizing the following four metrics:*

$$NSE_X = 1 - \frac{\sum_{i=1}^{N}(X_{est}-X_{obs})^2}{\sum_{i=1}^{N}(X_{obs}-\overline{X_{obs}})^2}, \tag{9}$$

$$R_X = \frac{\sum_{i=1}^{N}(X_{est}-\overline{X_{est}})(X_{obs}-\overline{X_{obs}})}{\sqrt{\sum_{i=1}^{N}(X_{est}-\overline{X_{est}})^2 \times \sum_{i=1}^{N}(X_{obs}-\overline{X_{obs}})^2}}, \tag{10}$$

$$RMSE_X = \sqrt{\frac{\sum_{i=1}^{N}(X_{est}-X_{obs})^2}{N}}, \tag{11}$$

$$Bias_X = \frac{\sum_{i=1}^{N}(X_{est}-X_{obs})}{N \times \overline{X_{obs}}}, \tag{12}$$

*where $NSE$, $R$, $RMSE$, and $Bias$ are the Nash-Sutcliffe Efficiency, the correlation coefficient, the Root Mean Square Error, and the ratio of the difference between the estimated and the observed to the observed average. The subscript X represents ET or GPP; the subscripts est and obs stand for the estimated and the observed, respectively.*

References here are the same as those in the manuscript.

---

## Author Comment (AC3)

**Reply to Reviewer 3**

**General Comments:**

1. This study used the PML-V2 model to develop ET and GPP datasets in China. The PML-V2 is calibrated and validated based on the data from 26 eddy covariance flux towers. The GPP and ET data developed in this study are compared with other global ET and GPP products and water balance data at the regional level. This study did a good job on model validation, but there still exist some issues in this stage.

**Response:**

Thank you for your very positive overall evaluation of the manuscript. We have carefully checked and re-edited it. In the following, we reply to all comments in a point-by-point response. All comments are shown in blue. Sentences from the manuscript are in italics and the revised contents are indicated in red.

2. As a data description paper, the methodology is an important section to let the audience know how the data is developed. However, the model description is not very clear and well organized in this paper. Although the PML-V2 model is already described in other papers, I think more details are still needed and could be put in the supplementary. Whether the code of the model is open source? If yes, a link to the model program should be provided. Why the PML-V2 (China) can simulate daily scale data while PML-V2 (Global) cannot? Are there any improvements in the model?

**Response:**

All the details of the PML-V2 model have now been reorganized and put in the supplement part. The PML-V2(China) source code is available through the public GitHub repository (https://github.com/SylviaHeee/PML-V2-China). First, PML-V2(China) uses a new parameter set for the country-wide simulation based on the daily EC observed, while PML-V2 (Global) uses the global parameters resolution that performs not well compared to PML-V2(China) at the plot scale and also at the basin-scale, shown in this manuscript. Second, PML-V2(China) uses daily input data while PML-V2 (Global) uses those at the 8-day scale. Third, the country-specific meteorological forcing, i.e., the China Meteorological Forcing Dataset (CMFD), is used to drive the PML-V2 in China, which is more accurate than those forcings extracted from global forcing products. Fourth, PML-V2(China) uses land surface temperature data, ERA5-Land, as the input surface temperature instead of air temperature like PML-V2 (Global) choosing to calculate the outgoing longwave radiation. Firth, PML-V2(China) utilizes the MODIS leaf area index data after the improved Whittaker filter and reveals the characteristics of the planting system.

3. The authors claimed the PML-V2 model performed better than other products. The evidence of the high accuracy of the ET and GPP mainly comes from the validation results at 26 EC sites. The 26 EC sites were used to calibrate and validate the model, while other global products did not calibrate and validate based on the same EC sites. If the PML-V2 and other products were used to compare against other new EC sites (not the 26 sites), can it still be the best one? It seems a little bit unfair to claim that this dataset is better than others when other models cannot

access these EC data. I encourage authors to also publish these EC data that are used in validation.

**Response:**

Here we use the water-balance ET to compare the accuracy of several products and it can be regarded as an independent validation since the PML-V2 (China) is not calibrated against the water-balance ET. The PML-V2 (China) model estimates get the smallest Bias of 6.28% and the highest NSE of 0.82 against water-balance annual ET estimates across 10 major river basins in China among five ET products.

The copyright of the EC data used in this study belongs to principal investigators of the EC stations. Therefore, we have no right to publish them. The EC data are all free to access from platforms on the Internet and their download links are provided in Table 2 and the references part of the manuscript.

4. According to the distribution of 26 EC sites (Fig 2), most of them are located in arid regions where ET may be low. The total estimated ET in China may be controlled by the ET estimated in the south region where few EC sites are located. There may exist large uncertainties in quantifying total ET.

**Response:**

We used the 26 EC-observed classified according to various plant function types (PETs) to get the 11 calibrated parameters for each PFTs, not based on climate types. Every PFTs have no less than one EC site. Besides, China has one of the largest dryland areas worldwide about 6.6 million km2 which covers 68.8% of the country (Prăvălie, 2016; Li et al., 2021). This shows the rationality of using more EC sites in arid areas. In addition, we have reselected the color ramps for the aridity index (AI) map based on United Nations Convention to Combat Desertification (as shown in Fig. R1 below), because the original AI colors between 0.6 and 1 were set as different degrees of yellow which may mislead readers that China has too much dryland (Fig 2 in the original manuscript).

Figure R1: Geographical locations of 26 EC flux towers for nine major IGBP PFTs, the main rivers, and the ten major river basins in China. Overlain are 20-year mean annual aridity index (AI) values during 2001-2020 using GLDAS-2.1, that is, the ratio of annual precipitation to Penman potential evapotranspiration. PFTs shown in legend are ENF (Evergreen Needleleaf Forests), EBF (Evergreen Broadleaf Forests), MF (Mixed Forests), OSH (Open Shrublands), SAV (Savannas), GRA (Grasslands), WET (Permanent Wetlands), CRO (Croplands), and BSV (Barren Sparse Vegetation).

**References**

Li, C., Fu, B., Wang, S., Stringer, L. C., Wang, Y., Li, Z., Liu, Y., and Zhou, W.: Drivers and impacts of changes in China's drylands, Nat Rev Earth Environ, 2, 858–873, https://doi.org/10.1038/s43017-021-00226-z, 2021.

Prăvălie, R.: Drylands extent and environmental issues. A global approach, Earth-Science Reviews, 161, 259–278, https://doi.org/10.1016/j.earscirev.2016.08.003, 2016.

5. In the discussion section, two advantages of this new dataset are provided, one is the watercarbon coupled process, and the other is more EC data help constrain the parameters. How are water and carbon coupled in the model? And why does the coupled carbon process help advance the model? There are many land surface models that couple water and carbon processes, but it is not always the case that these models performed better in simulating ET.

**Response:**

PML-V2 adopted coupling a photosynthesis model (Farquhar et al., 1980) and an improved canopy stomatal conductance model (Yu et al., 2004) with the Penman-Monteith (P-M) equation to estimate GPP and transpiration from the plant canopy ( $E_c$ ) collectively (Gan et al., 2018). Detailed descriptions of PML-V2 have been provided in the revised supplement. The most important fact is that  $E_c$  and GPP processes should be coupled through stomata. Not coupling these two processes can cause the following issues: (i) internal inconsistency between ET and GPP estimates if their forcing data are not the same; (ii) inaccurate causality analysis for mean annual values, trends/variation, and water use efficiency. To better understand the influence of carbon-constrained impacts on evapotranspiration, it is critical to credibly couple ET and GPP products at moderate spatial resolution (Zhang et al., 2019; Ma et al., 2022). PML-V2 is a water-carbon coupled but a parsimonious model with only 11 parameters. For most land surface models, they contain much more parameters that are hard to calibrate, which may cause uncertainties in simulating ET.

**References**

Ma, N. and Zhang, Y.: Increasing Tibetan Plateau terrestrial evapotranspiration primarily driven by precipitation, Agricultural and Forest Meteorology, 317, https://doi.org/10.1016/j.agrformet.2022.108887, 2022.

Zhang, Y., Kong, D., Gan, R., Chiew, F. H. S., McVicar, T. R., Zhang, Q., and Yang, Y.: Coupled estimation of 500 m and 8-day resolution global evapotranspiration and gross primary production in 2002–2017, Remote Sensing of Environment, 222, 165–182, https://doi.org/10.1016/j.rse.2018.12.031, 2019.

6. The daily data is one important advantage of this dataset. But there are no details of how daily data is better than the data at the 8-day scale.

**Response:**

Here we emphasized the advantage and implications of daily data against the data at the 8day scale is an improvement of temporal resolution of PML-V2(China) compared to current mainstream products with higher simulation accuracy. The cross-validated statistical indicators of the daily GPP estimated by PML-V2(China) at 26 EC flux towers have been added in Table 3. It is evident that PML-V2(China) at a daily scale excels its global version at the 8-day scale, rendered by *RMSE* being 0.48 mm d-1 lower for ET, and 1.30 g C m-2 d-1 lower for GPP, *NSE* being 0.04 higher for ET and 0.08 higher for GPP and *R* being 0.04 higher for ET and 0.05 higher for GPP. Then in the 4.3 section, we discussed the implications of the daily products. For instance, daily outputs from PML-V2(China) can be better used by the agricultural and water sectors for operational applications. Timely access to daily data at the regional or national scale helps the Ministry of Agriculture and Water Resources to develop better policies. Indeed, there is a remarkable relationship between soil water content and ET (Graf et al., 2014; Brust et al., 2021), so getting daily ET information accurately is of great significance for soil water depletion assessment, irrigation system design, and water resources management in agricultural areas, such as in the North China Plain.

Table 3: Statistical indicators of PML-V2(China) and other models for simulating ET and GPP at 26 EC flux towers. NSE and R values are unitless. The unit of RMSE for ET is mm  $d^{-1}$  while it is g C  $m^{-2} d^{-1}$  for GPP. The unit of Bias is %.

| Scale | Variable | Models         | NSE   | R    | RMSE | Bias   |
|-------|----------|----------------|-------|------|------|--------|
| daily | ET       | PML-V2(China)  | 0.66  | 0.84 | 0.33 | -7.97  |
|       |          | GLEAM          | 0.44  | 0.69 | 1.04 | -14.45 |
|       |          | SEBAL          | -7.10 | 0.16 | 3.95 | 5.31   |
| 8-day | ET       | PML-V2(China)  | 0.74  | 0.87 | 0.66 | -11.54 |
|       |          | PML-V2(Global) | 0.62  | 0.80 | 0.81 | -5.05  |
|       |          | MOD16A2        | 0.37  | 0.63 | 1.07 | -10.90 |
| daily | GPP      | PML-V2(China)  | 0.76  | 0.87 | 0.87 | -0.82  |
| 8-day | GPP      | PML-V2(China)  | 0.75  | 0.87 | 1.93 | -6.51  |
|       |          | PML-V2(Global) | 0.68  | 0.82 | 2.17 | -1.74  |
|       |          | MOD17A2H       | 0.49  | 0.78 | 2.74 | -38.79 |
|       |          | EC-LUE         | -0.04 | 0.35 | 3.91 | -41.91 |
|       |          | VPM            | 0.21  | 0.60 | 3.41 | -8.21  |

**Specific Comments:**

1. L60. The 8-day scale data is enough to detect seasonal changes.

**Response:**

We agree with you. The sentence has been modified and an appropriate reference has also been added below.

For instance, products with low temporal resolutions are erratic to detect subtle seasonal changes in areas seriously affected by human activities and in arid regions, such as irrigated farmland with a dry climate (Bodner et al., 2015) and an evergreen broad-leaf Mediterranean forest during severe summer drought (Liu et al., 2015).

**References**

Bodner, G., Nakhforoosh, A., and Kaul, H.-P.: Management of crop water under drought: a review, Agron. Sustain. Dev., 35, 401–442, https://doi.org/10.1007/s13593-015-0283-4, 2015.

Liu, J., Rambal, S., and Mouillot, F.: Soil Drought Anomalies in MODIS GPP of a Mediterranean Broadleaved Evergreen Forest, Remote Sensing, 7, 1154–1180, https://doi.org/10.3390/rs70101154, 2015.

2. L68-70. Whether this dataset has a better performance in simulating WUE. Different data sources of GPP and ET do not necessarily mean high uncertainties. If a water-carbon model is used to estimate GPP and ET, other information such as nutrient limitation may be lost, therefore, the estimated GPP may not be more accurate than directly observed data.

**Response:**

Figure R2 summarizes PML-V2 performance when estimating annual WUE for whole ecosystems at the 95 global flux sites, in comparison to other model performance, i.e., FluxCom GPP/GLEAM ET, VPM GPP/GLEAM ET, and MOD17 GPP/MOD16 ET (Zhang et al., 2019). PML-V2 performs reasonably well in estimating annual total WUE, indicated by the statistical metrics: NSE = 0.48,  $R^2 = 0.49$ , RMSE = 0.86 g C mm-1 H2O, Bias = 3.3%. Furthermore, PML-V2 is much better than the combinations of other products for estimating ecosystem WUE. This result indicates the benefit of using the coupled PML-V2 model for estimating ecosystem WUE as the use of the coupled GPP/ET models avoids internal inconsistencies between independent ET and GPP models which provides strong motivation for this research.

We agree that the estimated GPP may not be more accurate than directly observed data. But there are sparse and short-period ground observations in China, continuous gridded GPP products are needed to understand the spatial and temporal pattern of GPP. The paragraph from lines 68 to 70 has a chief sentence at first: *secondly, the phenomenon of ignoring the water-carbon coupling process frequently appearing in the existing products has brought systematic errors.* So, in this paragraph, we just introduced a problem in the existing ET and GPP gridded models. Moreover, the nutrient limitation for estimating GPP and ET deserves further study in the future.

---

## Author Response (AR1)

**Reply to Reviewers and Editor**

Dear editorial board members and reviewers,

The authors would like to thank the editors and three reviewers reviewing this manuscript. The authors sincerely thank the reviewers' insightful and constructive comments and suggestions. We appreciate your time and effort in considering this manuscript for publication. According to your nice suggestions, we have checked carefully and incorporated these comments to the revised version. Follows are the key changes in this submission:

- Bias correction of meteorological forcing inputs during 2019-2020 and remove the spatial and trend analysis in that period because the CMFD and the GLDAS-2.1 are different forcing datasets for driving the PML-V2 model;
- 2. The spatial analysis showing the ability of the PML-V2(China) product to identify the crop phenology has been added in the revised version;
- 3. The calibration using the genetic algorithm has been supplemented in materials and methods part;
- 4. The details of the PML-V2 model has been well-organized and put in the supplement; and
- 5. Minor issues have been corrected according to the comments from the reviewers and editors.

We reply to every comment point-by-point in the response letter. All comments are shown in blue. Sentences from the manuscript are in italics and the revised contents are indicated in red. We believe that the concerns from the reviewers have been addressed. Please let us know if there are any questions and queries. Thanks again for the editors and the reviewers for their valuable time, suggestions and comments.

**Editor**

**Comments:**

The study did a generally good job in producing GPP and ET simultanesly over China. After reading through the reviewers' comments and the authors' response, I feel there are still two issues needing to be resolved:

The first is the inconsistency between GLDAS and CMFD meteorological data. If the authors would like to publish the GPP and ET products during 2019-2020, I would suggest to at least bias-correct GLDAS data in the period using CMFD data before 2018 as the differences in GPP and ET produced by the two datasets are considerable.

The second is that the authors should avoid statements that PMLv2 performs better than other models. As reviewers pointed out, the results that PMLv2 derived GPP and ET products showed better performance may arise from its calibration using observations at 27 flux sites, which are maybe inaccessble to other models. Meanwhile, I cannot agree to conclude that PMLv2(China) performs better than PMLv2(Global) simply because the former runs using daily inputs but without process improvements. I suggest that the authors

only state the GPP and ET products produced in this study are better than other currently availabe data of the same type.

**Response:**

We appreciate your thoughtful and positive comments on our work. With the help of your constructive suggestions, we believe that this manuscript will be improved substantially. Following are our responses to your two questions:

- Bias correction of GLDAS data in 2019-2020. Yes, we agree that using bias correction of GLDAS forcings can eliminate the subsequent bias in estimating ET and GPP. After a comprehensive comparison of various bias correction methods, a widely used methodology, delta change (i.e., DC, also called change factor methodology), was selected in this study (Anandhi et al., 2011; Teutschbein and Seibert, 2012; Rasmussen et al., 2012; Hempel et al., 2013; Beck et al., 2018; Haro-Monteagudo et al., 2020). The underlying idea of the DC method is to use simulated future anomalies (i.e., GLDAS-2.1 in this study) for a perturbation of observed data (i.e., CMFD) rather than to use the simulations of future conditions directly. For each grid cell, we bias-corrected the daily meteorological data during 2019-2020 by monthly scaling factors. The details for bias correction have also been added to the manuscript.
- Internal comparison of PML-V2 versions. We agree that it is not appropriate to claim that PML-V2(China) is better than PML-V2(Global) simply because the former runs using daily local inputs but without process improvements. As such, we have changed the description of model performance in this revision based on your suggestions and the reviewers' comments.

**References**

- Anandhi, A., Frei, A., Pierson, D. C., Schneiderman, E. M., Zion, M. S., Lounsbury, D., and Matonse, A. H.: Examination of change factor methodologies for climate change impact assessment, Water Resources Research, 47, W03501, https://doi.org/10.1029/2010WR009104, 2011.
- Beck, H. E., Zimmermann, N. E., McVicar, T. R., Vergopolan, N., Berg, A., and Wood, E. F.: Present and future Köppen-Geiger climate classification maps at 1-km resolution, Sci Data, 5, 180214, https://doi.org/10.1038/sdata.2018.214, 2018.
- Haro-Monteagudo, D., Palazón, L., and Beguería, S.: Long-term sustainability of large water resource systems under climate change: A cascade modeling approach, Journal of Hydrology, 582, 124546, https://doi.org/10.1016/j.jhydrol.2020.124546, 2020.
- Hempel, S., Frieler, K., Warszawski, L., Schewe, J., and Piontek, F.: A trend-preserving bias correction – the ISI-MIP approach, Earth Syst. Dynam., 4, 219–236, https://doi.org/10.5194/esd-4-219-2013, 2013.
- Rasmussen, J., Sonnenborg, T. O., Stisen, S., Seaby, L. P., Christensen, B. S. B., and Hinsby, K.: Climate change effects on irrigation demands and minimum stream discharge: impact of bias-correction method, Hydrology and Earth System Sciences, 16, 4675– 4691, https://doi.org/10.5194/hess-16-4675-2012, 2012.
- Teutschbein, C. and Seibert, J.: Bias correction of regional climate model simulations for hydrological climate-change impact studies: Review and evaluation of different

| methods,       | Journal          | of         | Hydrology,    | 456–457, | 12–29, |
|----------------|------------------|------------|---------------|----------|--------|
| https://doi.or | g/10.1016/j.jhyc | lrol.2012. | 05.052, 2012. |          |        |

**Reviewer 1**

**General Comments:**

This study with a title of "A daily and 500m coupled evapotranspiration and gross primary production product across China during 2000-2020" has been seriously reviewed. Overall, this paper is well organized, including written English, structures, and the conclusions. Importantly, I believe that the PML-V2(China) product could provide a great opportunity for academic communities and various agencies for scientific studies and applications. However, before acceptance the authors should give the reasonable explanations to the following questions. So, I would like to recommend this paper to be conducted a major revision.

**Response:**

We appreciated tremendously your thoughtful comments and positive review on our article. According to your nice suggestions, we have checked and re-edited the original manuscript carefully. In the following, we reply to all comments in a point-by-point response. All comments are shown in blue. Sentences from the manuscript are in italics and the revised contents are indicated in red.

**Specific Comments:**

1. In the section 2.2.2, I found that the different meteorological forcings were used here, i.e., CMFD during 2000 to 2018, but GLDAS during 2019 to 2020. Although the authors compare the difference between PML-V2(China)GLDAS-2.1 and PML-V2(China)CMFD at the national scale. However, the author did not compare the liner trends of these simulations. Maybe, the authors could add the evaluations of the linear trends of ML-V2(China)GLDAS-2.1 and PML-V2(China)GLDAS-2.1 and PML-V2(China)GLDAS-2.1 and PML-V2(China)GLDAS-2.1 and PML-V2(China)GLDAS-2.1 and PML-V2(China)CMFD GPP and ET during 2000-2018 at different spatial scales (i.e., grid and national scales). Mainly because this product has a great potential to use for study the linear trends of GPP and ET by the scholars.

**Response:**

The China Meteorological Forcing Dataset (CMFD) was constructed by merging in situ measurements at 753 China Meteorological Administration stations with advanced retrospective analyses data from five remoting sensing or reanalysis data including Global Land Data Assimilation System (GLDAS) (He et al., 2020). We chose to use the CMFD dataset as meteorological inputs, because it shows much more accuracy and superior quality than other meteorological datasets in China, such as GLDAS meteorological data (He et al., 2020). We compared the magnitude and variability of the products using different meteorological forcing inputs, i.e., PML-V2(China)GLDAS-2.1 and PML-V2(China)CMFD, at the grid and national scale in section 4.4.2 of the first draft, as follows:

To extend the simulation period, we used GLDAS-2.1 meteorological forcing data during 2019-2020 since the CMFD dataset is only up to 2018. To check if using these two datasets generates a systematic bias, we reran the PML-V2(China) in 2001-2018 using GLDAS-2.1 and compared the modelling results with those obtained using CMFD (Fig. S1). At the

national scale, the mean difference, calculated by  $(PML-V2(China)_{GLDAS-2.1} - PML-V2(China)_{CMFD})/PML-V2(China)_{CMFD}$ , varied from -1.22% to 1.62% among  $E_i$ ,  $E_c$ , and GPP, and was 13.72% for  $E_s$  and 7.78% for ET. The difference is within -25% ~ 25% in more than 66% of the research region for all five variables (Fig. S1b2-e2), specifically 100% for GPP, 95% for  $E_c$ , 84% for ET, 73% for  $E_i$ , and 66% for  $E_s$  (Fig. S1b3-e3). This illustrates that PML-V2(China) using the GLDAS-2.1 in 2019-2020 does not generate a noticeable systematic deviation.

The PML-V2(China) product of 2019-2020 is the interim data as the supplement of PML-V2(China) after 2018. We suggest that users do spatial variability analysis instead of trend analysis if they want to use the PML-V2(China) from 2019 to 2020. With the release of the meteorological dataset, we will continue to update the PML-V2(China) using the CMFD inputs. Moreover, we have removed the description and figures about the trend analysis of PML-V2(China) for 2019-2020 in the manuscript.